# STDP as Probabilistic Attribution: An Exact-Balance Continuous Kernel for Normalized Temporal Credit Assignment

## Abstract

We introduce a unified continuous kernel for spike-timing-dependent plasticity (STDP) that connects local spike-timing updates to normalized probabilistic attribution in convergent circuits. Classical phenomenological STDP models describe long-term potentiation (LTP; the strengthening of synapses when a presynaptic spike precedes a postsynaptic spike) and long-term depression (LTD; the weakening of synapses in the reverse order) using piecewise timing windows, while standard simulator implementations commonly realize such rules with local traces. Our contribution is therefore not a claim of asymptotic speedup over trace-based STDP, but a single differentiable trace-interaction kernel whose induced learning window can be analyzed in closed form. The model represents presynaptic and postsynaptic events by dimensionless exponentially decaying traces and defines synaptic change by their cooperative and competitive interaction. For an isolated pre-post spike pair, we derive the closed-form STDP window and prove that the total integrated potentiation and depression areas are exactly balanced for all positive decay rates. We further summarize parameter sweeps and component ablations showing how the two decay rates tune window morphology and why both the multiplicative gating term and competitive difference term are required for a biphasic timing-sensitive window. A fit to the classical data of Bi & Poo (1998) gives $R^2 = 0.63$ and reveals a narrow near-coincident positive-update regime for small post-before-pre lags. This regime is a structural consequence of fitting a continuous kernel with mismatched decay rates; however, the raw observations in the corresponding interval are positive, so we treat it as a data-consistent near-synchronous attribution hypothesis rather than a confirmed biological mechanism or a mere fitting artifact. Forcing the zero crossing to $\Delta t = 0$ substantially worsens the fit (Appendix D). At the network level, we show that when the additive kernel is combined with multiplicative afferent normalization, the mean-field dynamics reduce to a delta-rule-like update whose fixed point is a normalized event-rate target, $w_i^* = \nu_i q_i / \sum_j \nu_j q_j$. Under a strict causal-window approximation, $q_i = P(\text{Post} \mid \text{Pre}_i)$, and under the corresponding partition assumptions this target can be interpreted as posterior attribution, $P(\text{Pre}_i \mid \text{Post})$. Without those assumptions, the fixed point should be read as normalized conditional event-rate attribution. Simulations confirm convergence in sparse regimes, document progressive degradation under dense firing, and show that the proposed kernel outperforms classical STDP baselines, including variants matched for area balance, trace mode, step size, and temporal footprint, under identical normalization. An iso-rate control confirms the network tracks attribution probability independent of firing rate, a decorrelated control confirms it tracks the product $\nu_i q_i$, and a heterogeneous-delay condition confirms robustness to non-uniform causal timing.

## 1 Introduction

### 1.1 Background

Spike-timing-dependent plasticity (STDP) is a temporally sensitive form of synaptic plasticity in which synaptic change depends on the relative timing of presynaptic and postsynaptic spikes. In the classical

asymmetric window, a presynaptic spike followed shortly by a postsynaptic spike tends to potentiate the synapse, while the reverse ordering tends to depress it (Markram et al., 1997; Bi & Poo, 1998; Caporale & Dan, 2008; Feldman, 2012). STDP has therefore served as a central model for how local neural events can shape circuit organization, sequence sensitivity, and credit assignment over time.

Many phenomenological STDP models describe potentiation and depression using separate, piecewise branches of a temporal learning window (Gerstner et al., 1996; Song et al., 2000; Pfister & Gerstner, 2006). These descriptions are useful and biologically grounded, and modern spiking-network simulators commonly implement them efficiently with local exponentially decaying traces rather than naive all-pairs spike comparisons (Morrison et al., 2008; Gerstner et al., 2014). The motivation for the present work is therefore not that classical STDP lacks efficient trace implementations. Instead, the goal is to provide a single continuous interaction function whose closed-form properties can be studied directly and whose relationship to normalized probabilistic attribution can be made explicit.

## 1.2 Related Work and Scope

The proposed kernel sits between two literatures. The first is the phenomenological STDP literature, which characterizes pair-based, triplet-based, and rate-dependent plasticity windows (Gerstner et al., 1996; Song et al., 2000; Pfister & Gerstner, 2006; Morrison et al., 2008). The second is the probabilistic and normative plasticity literature, which has shown that STDP-like rules combined with competition, normalization, or stochastic synaptic dynamics can implement forms of Bayesian inference, expectation-maximization, hidden-state learning, and synaptic sampling (Nessler et al., 2013; Kappel et al., 2014; 2015; Aitchison et al., 2021). This paper does not claim that the idea of probabilistic STDP is new. Rather, it contributes a compact continuous kernel whose isolated-pair window is analytically solvable, whose total area is exactly balanced without tuning separate LTP and LTD amplitudes, and whose normalized mean-field fixed point can be written transparently in terms of event-rate attribution.

## 1.3 Current Contribution

We make three main theoretical claims and add two supporting analyses. First, a cubic interaction between dimensionless presynaptic and postsynaptic traces produces a continuous STDP window without explicit LTP/LTD branching in the governing equation. Second, the induced isolated-pair window has exactly zero integrated area for all positive decay rates. This is a structural property of the kernel, not a parameter-fitting constraint. Third, when the additive local rule is embedded in a convergent circuit with multiplicative afferent normalization, the mean-field dynamics converge to normalized kernel-weighted attribution rates. Under additional causal-window and event-partition assumptions, these normalized rates can be interpreted as posterior credit assignment. The supporting analyses quantify how decay rates shape the learning window and use component ablations to show that both multiplicative overlap gating and competitive trace comparison are needed for the full biphasic rule.

A point of scope is important. The base kernel is additive and weight-independent; by itself it is not a bounded single-synapse estimator of $P(\text{Post} \mid \text{Pre}_i)$. Stability and competition enter through the afferent normalization analyzed below. We therefore omit the previous single-synapse fixed-point argument and treat the normalized convergent circuit as the primary probabilistic object.

In machine learning terms, the normalized fixed point can be interpreted as local credit assignment among competing afferents: it estimates which input streams most reliably account for a postsynaptic event. In a multi-neuron layer, such a mechanism could support competitive clustering, prototype assignment, or local representation learning without backpropagation. The present experiments are not task benchmarks; they are unit tests of the statistical fixed point under controlled event statistics. The open-loop simulation setup isolates the statistical target predicted by the mean-field theory; it does not claim closed-loop task learning, because weights do not drive postsynaptic spikes in these verification experiments. Functional benchmarks and closed-loop evaluations are future work.

## 2 Unified Continuous STDP Kernel

### 2.1 Dimensionless Event Traces

Let $x_i(t)$ denote the presynaptic trace associated with input $i$, and let $y(t)$ denote the postsynaptic trace. These traces are dimensionless eligibility variables, not literal measurements of membrane voltage. The notation $r_E$ and $r_A$ is retained to indicate the decay rates associated with presynaptic events (historically termed excitatory postsynaptic potentials, or EPSPs) and postsynaptic events (action potentials, or APs), respectively; however, the traces $x_i$ and $y$ are abstract and unitless.

For a single isolated presynaptic event at time $t_{\text{pre}}$ and a single postsynaptic event at time $t_{\text{post}}$, the traces are

$$x_i(t) = \begin{cases} 0, & t < t_{\text{pre}}, \\ e^{-r_E(t-t_{\text{pre}})}, & t \geq t_{\text{pre}}, \end{cases} \tag{1}$$

$$y(t) = \begin{cases} 0, & t < t_{\text{post}}, \\ e^{-r_A(t-t_{\text{post}})}, & t \geq t_{\text{post}}. \end{cases} \tag{2}$$

For a spike train, the version analyzed here uses nearest-spike hard-reset traces. Between events,

$$\frac{dx_i}{dt} = -r_E x_i, \qquad \frac{dy}{dt} = -r_A y, \tag{3}$$

and at event times,

$$x_i(t_{\text{pre},i}^+) = 1, \qquad y(t_{\text{post}}^+) = 1. \tag{4}$$

Equivalently, $x_i(t)$ and $y(t)$ are Markovian trace states determined by the most recent relevant event and the time since that event. This nearest-spike approximation is analytically convenient and is distinct from additive trace models, in which repeated spikes accumulate trace amplitude. The implications of hard resets for bursty or dense spike trains are treated as a limitation and as a simulation target below.

### 2.2 Trace-Interaction Kernel

Synaptic change at input $i$ is defined by

$$\frac{dw_i}{dt} = \eta \, y(t) \, x_i(t) \, \big( y(t) - x_i(t) \big), \tag{5}$$

where $\eta > 0$ is a scale or learning-rate parameter. The product $y(t)x_i(t)$ is cooperative: both traces must be nonzero for plasticity to occur. The difference $y(t)-x_i(t)$ is competitive: the sign of the update is determined by which trace is larger. When the postsynaptic trace dominates, the instantaneous update is positive; when the presynaptic trace dominates, it is negative.

This equation is continuous and differentiable in the trace variables. It should be read as an analytically unified local plasticity kernel, not as a claim that all classical trace-based STDP implementations are computationally inefficient. The value of the formulation is that it yields a closed-form isolated-pair window and an exact balance theorem without separately specifying LTP and LTD branches.

## 3 Single-Pair Window and Exact Balance

### 3.1 Alignment with Empirical STDP Data

To compare the induced learning window with the classical data of Bi & Poo (1998), define the inter-spike interval

$$\Delta t = t_{\text{post}} - t_{\text{pre}}. \tag{6}$$

A positive $\Delta t$ denotes pre-before-post timing, while a negative $\Delta t$ denotes post-before-pre timing.

For each candidate pair $(r_E, r_A)$, the raw predicted window $\Delta w_{\mathrm{raw}}(\Delta t; r_E, r_A)$ was computed from the closed-form expressions below. Because the empirical data and model window are expressed in different arbitrary amplitude units, the predicted window was vertically rescaled by a scalar $c$:

$$\Delta w(\Delta t; r_E, r_A, c) = c\, \Delta w_{\mathrm{raw}}(\Delta t; r_E, r_A). \tag{7}$$

For each $(r_E, r_A)$, $c$ was chosen to minimize

$$\mathrm{SSD}(r_E, r_A, c) = \sum_j \left[ \Delta w_{\mathrm{emp},j} - c\, \Delta w_{\mathrm{raw}}(\Delta t_{\mathrm{emp},j}; r_E, r_A) \right]^2. \tag{8}$$

The best fit in the current analysis is

$$r_E = 0.1782\,\mathrm{ms}^{-1}, \qquad r_A = 0.0775\,\mathrm{ms}^{-1}, \qquad c = 106.5064,$$

with $R^2 = 0.63$. The Bi and Poo data comprise $n = 60$ measurements digitized from Figure 1 of the original publication; the digitized values and fitting script are included in the supplementary code. Parameters were optimized by differential-evolution search minimizing Equation 8. The scalar $c$ is used only to place the theoretical window on the empirical plotting scale; it is distinct from the learning-rate scale $\eta$ in Equation 5 and from the mean-field increment $K$ introduced later.

## 3.2 Closed-Form STDP Window for an Isolated Spike Pair

For one presynaptic spike at $t_{\mathrm{pre}}$ and one postsynaptic spike at $t_{\mathrm{post}}$, the integrand in Equation 5 is nonzero only after both traces have been activated:

$$t \geq t_{\mathrm{start}} = \max(t_{\mathrm{pre}}, t_{\mathrm{post}}).$$

The net weight change for the pair is

$$\Delta w(\Delta t) = \eta \int_{t_{\mathrm{start}}}^{\infty} y(t) x_i(t) \big( y(t) - x_i(t) \big)\, dt. \tag{9}$$

For $\Delta t \geq 0$, set $\tau = t - t_{\mathrm{post}} \geq 0$. Then $y(t) = e^{-r_A \tau}$ and $x_i(t) = e^{-r_E(\tau + \Delta t)}$, yielding

$$\Delta w(\Delta t) = \eta \int_0^{\infty} \left[ e^{-2r_A \tau} e^{-r_E(\tau + \Delta t)} - e^{-r_A \tau} e^{-2r_E(\tau + \Delta t)} \right] d\tau \tag{10}$$

$$= \eta \left[ \frac{e^{-r_E \Delta t}}{2r_A + r_E} - \frac{e^{-2r_E \Delta t}}{r_A + 2r_E} \right], \qquad \Delta t \geq 0. \tag{11}$$

For $\Delta t < 0$, set $\tau = t - t_{\mathrm{pre}} \geq 0$. Then $y(t) = e^{-r_A(\tau - \Delta t)}$ and $x_i(t) = e^{-r_E \tau}$, yielding

$$\Delta w(\Delta t) = \eta \int_0^{\infty} \left[ e^{-2r_A(\tau - \Delta t)} e^{-r_E \tau} - e^{-r_A(\tau - \Delta t)} e^{-2r_E \tau} \right] d\tau \tag{12}$$

$$= \eta \left[ \frac{e^{2r_A \Delta t}}{2r_A + r_E} - \frac{e^{r_A \Delta t}}{r_A + 2r_E} \right], \qquad \Delta t < 0. \tag{13}$$

Equations 11–13 define the isolated-pair STDP window induced by the continuous kernel.

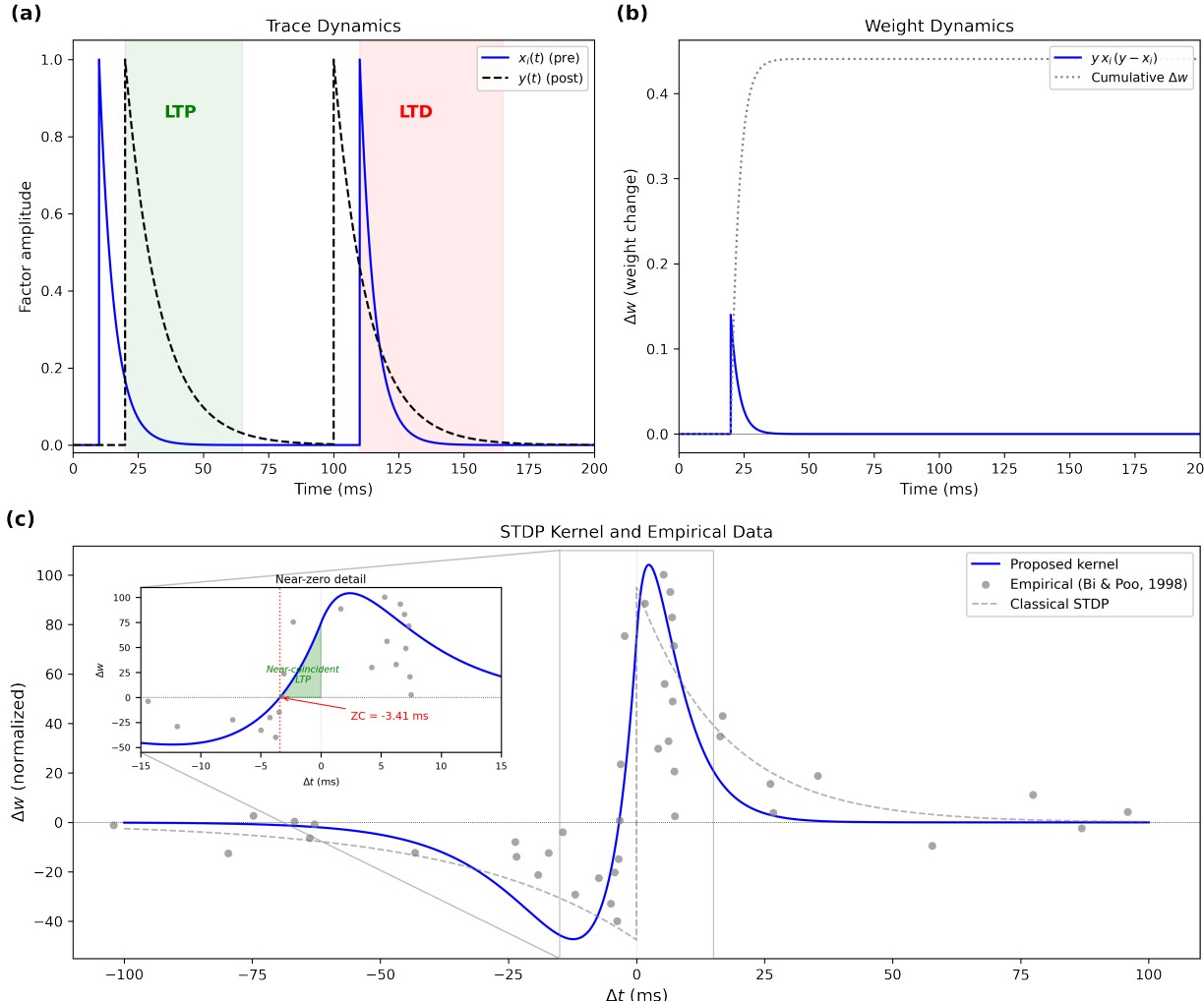

Figure 1: **(a)** Dimensionless presynaptic trace $x_i(t)$ (blue solid) and postsynaptic trace $y(t)$ (black dashed) for two example spike pairs. Left pair: pre-before-post ($\Delta t = +10$ ms), producing potentiation (green shading) because $y > x_i$ in the overlap region. Right pair: post-before-pre ($\Delta t = -10$ ms), producing depression (red shading) because $x_i > y$ in the overlap region. Traces undergo hard reset to 1 at each spike (Equation 4). **(b)** Instantaneous kernel value $y\,x_i\,(y - x_i)$ (blue) and cumulative weight change (gray dotted) for the pre-before-post pair in (a). The instantaneous signal peaks shortly after the postsynaptic spike and decays as both traces diminish. **(c)** STDP window generated by the fitted decay parameters ($r_A = 0.0775$ ms$^{-1}$, $r_E = 0.1782$ ms$^{-1}$) and compared with data adapted from Bi and Poo (1998). A classical piecewise STDP window (gray dashed) is shown for reference. *Inset:* near-zero detail showing the primary zero crossing at $\Delta t \approx -3.41$ ms (red dotted line). The green-shaded region marks the narrow near-coincident positive-update interval for small post-before-pre lags discussed in Section 3.3. The raw observations falling within this interval are positive, with no negative observations in the same region.

### 3.3 Near-Coincident Positive Updates and Temporal-Attribution Scope

The fitted rates satisfy $r_E > r_A$, which shifts the zero crossing of the negative-lag branch slightly to the left of zero. Setting Equation 13 to zero gives

$$\Delta t_0^- = \frac{1}{r_A} \log \left( \frac{2r_A + r_E}{r_A + 2r_E} \right).$$  (14)

For the fitted parameters above, $\Delta t_0^- \approx -3.4\,\text{ms}$. Thus, the continuous kernel assigns positive integrated weight change to a narrow interval of small post-before-pre lags, $\Delta t_0^- < \Delta t < 0$. Mechanistically, this occurs because immediately after a near-coincident presynaptic event, $x_i$ resets to 1 and can exceed the decayed postsynaptic trace $y$, producing an initial LTD component; however, since $x_i$ decays faster than $y$ (because $r_E > r_A$), the traces subsequently cross and the later LTP tail out-integrates the initial depression. The net effect for the full isolated pair is a small positive weight change.

We do not interpret this region as a mere fitting anomaly. In the raw Bi & Poo (1998) measurements, the observations falling within the fitted near-coincident interval $[-3.4, 0]$ ms are positive, with no negative observations in the same interval. Thus, the fitted kernel preserves a local empirical pattern that is suppressed by classical piecewise STDP models that force the sign change to occur exactly at $\Delta t = 0$. At the same time, the shifted zero crossing is fundamentally a kinetic consequence of fitting a continuous polynomial with mismatched decay rates: because $r_E > r_A$, the presynaptic trace decays faster than the postsynaptic trace in the tail of the interaction, and the resulting late potentiation phase outweighs the initial depression phase for a narrow band of near-coincident lags. We treat this effect as a data-consistent near-synchronous attribution regime and a hypothesis for further biological testing, not as definitive evidence for a separate anti-causal potentiation mechanism. A quantitative comparison against a constrained fit that forces the zero crossing to $\Delta t = 0$ is provided in Appendix D.

This feature affects the interpretation of the learned statistic. A strict causal interpretation treats positive evidence as pre-before-post timing and identifies the relevant event probability with $P(\text{Post} \mid \text{Pre}_i)$. The full fitted kernel is slightly broader: its positive attribution signal can include causal and near-coincident relationships. We therefore formulate the mean-field result below in terms of a kernel-weighted attribution probability $q_i$. Under a strict causal-window approximation, $q_i = P(\text{Post} \mid \text{Pre}_i)$. Under the full fitted kernel, $q_i$ can include near-coincident temporal association. The consequences of this broader regime are left for future work.

### 3.4 Intrinsic Balance between LTP and LTD

The closed-form window has an exact isolated-pair balance property. Define

$$A_{\text{total}} = \int_{-\infty}^{\infty} \Delta w(\Delta t) \, d\Delta t = A_- + A_+,$$

where $A_-$ and $A_+$ are the areas over negative and positive inter-spike intervals. Omitting the common factor $\eta$, Equation 13 gives

$$A_- = \left[ \frac{e^{2r_A \Delta t}}{2r_A(2r_A + r_E)} - \frac{e^{r_A \Delta t}}{r_A(r_A + 2r_E)} \right]_{-\infty}^{0},$$

and Equation 11 gives

$$A_+ = \left[ -\frac{e^{-r_E \Delta t}}{r_E(2r_A + r_E)} + \frac{e^{-2r_E \Delta t}}{2r_E(r_A + 2r_E)} \right]_{0}^{\infty}.$$

Evaluating the limits,

$$A_{\text{total}} = \left( \frac{1}{2r_A(2r_A + r_E)} - \frac{1}{r_A(r_A + 2r_E)} \right) + \left( \frac{1}{r_E(2r_A + r_E)} - \frac{1}{2r_E(r_A + 2r_E)} \right)$$  (15)

$$= \frac{1}{2r_A + r_E} \left( \frac{1}{2r_A} + \frac{1}{r_E} \right) - \frac{1}{r_A + 2r_E} \left( \frac{1}{r_A} + \frac{1}{2r_E} \right)$$  (16)

$$= \frac{1}{2r_A r_E} - \frac{1}{2r_A r_E} = 0.$$  (17)

Therefore,

$$\int_{-\infty}^{\infty} \Delta w(\Delta t) \, d\Delta t = 0, \qquad r_A, r_E > 0. \tag{18}$$

This means that the isolated-pair window has no built-in area bias toward net potentiation or net depression. Long-term drift in a network must therefore come from structured temporal correlations, normalization constraints, or deviations from the isolated-pair assumptions.

### 3.5 Scope of the Balance Result

Equation 18 is exact for isolated pairs integrated over the full decay tails. It should not be read as an unconditional guarantee that arbitrary dense spike trains produce exactly zero background drift. Under the hard-reset dynamics in Equation 4, a new spike can truncate a previous trace trajectory; under overlapping spike trains, the cubic interaction $y^2 x_i - y x_i^2$ also introduces higher-order trace statistics. We therefore distinguish the exact isolated-pair theorem from the sparse-spiking approximation used in the mean-field analysis.

A useful way to express this distinction is to define an uncorrelated-background drift term

$$B_i := \mathbb{E}_{\text{uncorr}} \left[ \eta \, y(t) x_i(t) \big( y(t) - x_i(t) \big) \right]. \tag{19}$$

For sparse spike trains whose relevant interactions are well approximated by isolated pairs, $B_i \approx 0$. For dense or bursty regimes, $B_i$ may be nonzero and should be measured or bounded. The numerical verification plan below treats this as a first-order robustness check rather than assuming it away.

### 3.6 Parameter Sensitivity and Component Ablations

The balance theorem above is analytical. We also use finite-window numerical sweeps and component ablations as supporting checks on the kernel's morphology and design rationale. These analyses are not simulations of the network-level fixed point; they characterize the isolated-pair learning window and the role of each factor in Equation 5. Full details are provided in Appendices A–C.

For the fitted kernel, the key isolated-window metrics are summarized in Table 1. The reported area ratio is a finite-window numerical estimate; the exact continuous result remains Equation 18.

Table 1: Key metrics of the fitted isolated-pair STDP window. Amplitudes are reported in the unscaled model units used for window-shape analysis; the empirical plotting scale is controlled separately by the scalar $c$ in Equation 7.

| Metric | Value |
|---|---|
| Presynaptic trace decay $r_E$ | $0.1782 \, \text{ms}^{-1}$ |
| Postsynaptic trace decay $r_A$ | $0.0775 \, \text{ms}^{-1}$ |
| Maximum potentiation $\text{LTP}_{\max}$ | $0.978$ at $\Delta t = 2.41 \, \text{ms}$ |
| Maximum depression $\text{LTD}_{\max}$ | $-0.443$ at $\Delta t = -12.34 \, \text{ms}$ |
| Dynamic range | $1.421$ |
| Peak magnitude ratio $\text{LTP}_{\max}/|\text{LTD}_{\max}|$ | $2.206$ |
| Finite-window area ratio $A_{\text{LTP}}/A_{\text{LTD}}$ | $0.9995$ |
| Primary zero crossing | $-3.41 \, \text{ms}$ |

A grid sweep over $r_A, r_E \in \{0.05, 0.1, 0.2, 0.4\} \, \text{ms}^{-1}$ shows that window morphology changes systematically with the relative trace timescales. Increasing $r_A$, which shortens the postsynaptic trace, reduces the peak LTP magnitude; increasing $r_E$, which shortens the presynaptic trace, reduces the peak LTD magnitude. When $r_A = r_E$, the LTP and LTD lobes become symmetric in peak morphology. Across the finite-window numerical sweep, estimates of $A_{\text{LTP}}/A_{\text{LTD}}$ cluster tightly around one, consistent with but not replacing the exact theorem.

Component ablations further clarify why the full cubic form is needed. The product $yx_i$ alone is a coincidence gate but is potentiation-only. The difference $y - x_i$ alone can determine sign, but without multiplicative gating it lacks timing-localized overlap and its full-tail pair integral depends only on the trace time constants. Saturation or removal of either factor likewise collapses the rule to one-directional plasticity. The full expression $yx_i(y - x_i)$ combines both properties: local overlap gating and a temporal comparator. These ablations support the kernel design without changing the main claim that network-level stability is supplied by normalization, not by an isolated bounded single-synapse rule.

## 4 Normalized Attribution in a Convergent Network

### 4.1 Setup and Assumptions

Consider a postsynaptic neuron receiving input from $N$ presynaptic sources indexed by $i = 1, \ldots, N$. Let $w_i(t)$ be the weight from input $i$, with afferent normalization enforcing

$$\sum_{k=1}^{N} w_k = 1. \tag{20}$$

This normalization is local to the postsynaptic cell's afferent weight vector. It is a mathematical abstraction of homeostatic competition or synaptic scaling, not a claim that biological circuits implement instantaneous exact division after every event (Turrigiano, 2017).

Let input $i$ emit learning-eligible presynaptic events at rate $\nu_i$. Let $q_i$ denote the probability that such an event produces a positive kernel-weighted attribution signal. In the strict causal-window approximation, $q_i = P(\text{Post} \mid \text{Pre}_i)$. In the full fitted kernel, $q_i$ can also include near-coincident positive-update events that occur at small negative lags, as discussed in Section 3.3. Define the attribution event rate

$$R_i := \nu_i q_i, \tag{21}$$

and

$$R_{\text{tot}} := \sum_{j=1}^{N} R_j = \sum_{j=1}^{N} \nu_j q_j. \tag{22}$$

Let $K > 0$ denote the average positive increment induced by one successful attribution event at the responsible synapse. This quantity bundles the learning-rate $\eta$, the integrated effect of the kernel over the relevant lag distribution, and the scale of the discrete approximation. We assume $K \ll 1$, so first-order approximations in $K$ are appropriate.

### 4.2 From Continuous Kernel to Discrete Attribution Events

The base kernel (Equation 5) is a continuous biphasic ODE that produces both potentiation and depression. The discrete mean-field model below uses only a positive increment $+K$. This subsection explains the connection.

For synapse $i$, the expected direct weight-change rate from the continuous kernel can be approximated, in the sparse-spiking limit where trace interactions are well described by isolated pairs, as

$$\mathbb{E}\left[\frac{dw_i^{\text{direct}}}{dt}\right] \approx \eta \int_{-\infty}^{\infty} W(\Delta t) \, G_i(\Delta t) \, d\Delta t. \tag{23}$$

This is a leading-order sparse-spiking approximation; overlapping traces and reset truncation introduce higher-order terms collected in the background drift $B_i$ (Equation 19). Here $W(\Delta t)$ denotes the unscaled isolated-pair window, i.e., Equations 11–13 with the common factor $\eta$ omitted, so that $\eta$ appears exactly once in the expression above. The cross-correlation intensity can be decomposed as

$$G_i(\Delta t) = G_i^0 + G_i^{\text{excess}}(\Delta t), \tag{24}$$

where $G_i^0$ is the uniform baseline expected from independent Poisson firing, and $G_i^{\text{excess}}(\Delta t)$ is the excess intensity due to causal and near-coincident temporal structure. The excess component can be factored as

$$G_i^{\text{excess}}(\Delta t) = R_i \, p_i^{\text{excess}}(\Delta t), \tag{25}$$

where $R_i = \nu_i q_i$ is the attribution-event rate and $p_i^{\text{excess}}(\Delta t)$ is a unit-mass lag distribution conditional on a successful attribution event.

By the exact area balance theorem (Equation 18), the integral of the learning window against the uniform baseline is

$$\eta \, G_i^0 \int_{-\infty}^{\infty} W(\Delta t) \, d\Delta t = 0. \tag{26}$$

This is the critical step: under the sparse isolated-pair approximation, uncorrelated background spikes yield an expected weight change of exactly zero because the potentiation and depression lobes of the kernel cancel. The surviving contribution comes only from the excess correlation:

$$\eta \int_{-\infty}^{\infty} W(\Delta t) \, G_i^{\text{excess}}(\Delta t) \, d\Delta t = R_i \, K_i, \tag{27}$$

with

$$K_i := \eta \int_{-\infty}^{\infty} W(\Delta t) \, p_i^{\text{excess}}(\Delta t) \, d\Delta t. \tag{28}$$

Here $K_i$ is the average weight increment per successful attribution event at synapse $i$, not an update rate. In the simulation setup below, the excess lag distribution is a delta function at the fixed causal delay $d = 5$ ms, so $p_i^{\text{excess}}(\Delta t) = \delta(\Delta t - d)$ and $K_i = \eta \, W(d)$. For simplicity, we approximate all $K_i \approx K$ when the causal delays are similar across inputs.

Thus, the LTD lobe of the STDP kernel is not discarded; it cancels the uncorrelated background under the sparse-spiking approximation, leaving only the causal excess $K$ as the net per-event increment. Depression in the network model arises from competitive normalization (Section 4.3), not from the direct LTD of the isolated-pair window.

### 4.3 Single-Event Dynamics under Normalization

First consider an attribution event assigned to input $i$. Before normalization, synapse $i$ is potentiated by $K$:

$$w_i \to w_i + K.$$

The total incoming weight becomes $1 + K$. Multiplicative normalization gives

$$w_i' = \frac{w_i + K}{1 + K},$$

so

$$\Delta w_i = \frac{w_i + K}{1 + K} - w_i = \frac{K(1 - w_i)}{1 + K} \approx K(1 - w_i).$$

Averaging over attribution events at input $i$,

$$\mathbb{E}[\Delta w_i \mid \text{Pre}_i] \approx q_i K(1 - w_i). \tag{29}$$

Now consider an attribution event assigned to some other input $j \neq i$. Synapse $j$ is potentiated by $K$, and normalization rescales synapse $i$:

$$w_i' = \frac{w_i}{1 + K}, \qquad \Delta w_i = -\frac{K}{1 + K} w_i \approx -K w_i.$$

Thus,

$$\mathbb{E}[\Delta w_i \mid \text{Pre}_j] \approx -q_j K w_i, \qquad j \neq i. \tag{30}$$

This negative term is not direct LTD at synapse $i$. It is competitive depression induced by normalization after potentiation at a different afferent synapse. This is why the additive base kernel need not contain explicit weight-dependent bounds: the $(1 - w_i)$ and $-w_i$ terms emerge from normalization.

### 4.4 Mean-Field Dynamics and Fixed Point

Weighting Equations 29-30 by event rates gives

$$\frac{dw_i}{dt} \approx \nu_i q_i K(1 - w_i) - K w_i \sum_{j \neq i} \nu_j q_j \tag{31}$$

$$= K\left[R_i(1 - w_i) - w_i(R_{\text{tot}} - R_i)\right] \tag{32}$$

$$= K\left[R_i - w_i R_{\text{tot}}\right] \tag{33}$$

$$= K R_{\text{tot}} \left(\frac{R_i}{R_{\text{tot}}} - w_i\right). \tag{34}$$

The fixed point satisfies $dw_i/dt = 0$, hence

$$w_i^* = \frac{R_i}{R_{\text{tot}}} = \frac{\nu_i q_i}{\sum_j \nu_j q_j}. \tag{35}$$

Under the strict causal-window approximation $q_i = P(\text{Post} \mid \text{Pre}_i)$, this becomes

$$w_i^* = \frac{\nu_i P(\text{Post} \mid \text{Pre}_i)}{\sum_j \nu_j P(\text{Post} \mid \text{Pre}_j)}. \tag{36}$$

If the presynaptic events form an appropriate mutually exclusive and exhaustive partition for the postsynaptic event, then $\nu_i P(\text{Post} \mid \text{Pre}_i)$ is proportional to $P(\text{Pre}_i \wedge \text{Post})$, and the denominator is proportional to $P(\text{Post})$. Under those additional assumptions,

$$w_i^* = \frac{P(\text{Pre}_i \wedge \text{Post})}{P(\text{Post})} = P(\text{Pre}_i \mid \text{Post}). \tag{37}$$

Without these partition assumptions, Equation 35 should be read more conservatively as a normalized conditional event-rate or kernel-weighted attribution target. If all presynaptic sources have equal event rates and the causal-window approximation holds, Equation 36 reduces to

$$w_i^* = \frac{P(\text{Post} \mid \text{Pre}_i)}{\sum_j P(\text{Post} \mid \text{Pre}_j)}. \tag{38}$$

### 4.5 Background Drift Correction

The derivation above isolates excess attribution events and assumes the uncorrelated background contribution is negligible. Using the drift term in Equation 19, a first-order normalized correction can be written as

$$\frac{dw_i}{dt} \approx K\left(R_i - w_i R_{\text{tot}}\right) + B_i - w_i \sum_{k=1}^{N} B_k. \tag{39}$$

When $B_i \approx 0$ for all inputs, Equation 39 reduces to the clean delta rule in Equation 34.

For independent Poisson pre- and postsynaptic spike trains with hard-reset traces, the background drift admits a closed-form expression. Let $\lambda_i = \nu_i/1000$ and $\lambda_{\text{post}}$ denote the firing rates in events per millisecond. The hard-reset trace $x_i$ has the stationary moments $\mathbb{E}[x_i^n] = \lambda_i/(\lambda_i + n r_E)$, and similarly for $y$. Under the independence assumption, the expected instantaneous kernel value is $\mathbb{E}[y^2 x_i] - \mathbb{E}[y x_i^2] = \mathbb{E}[y^2]\mathbb{E}[x_i] - \mathbb{E}[y]\mathbb{E}[x_i^2]$, giving

$$B_i = \eta \, \lambda_{\text{post}} \lambda_i \cdot \frac{\lambda_{\text{post}} \, r_E - r_A \, \lambda_i}{(\lambda_{\text{post}} + 2r_A)(\lambda_i + r_E)(\lambda_{\text{post}} + r_A)(\lambda_i + 2r_E)}. \tag{40}$$

In the sparse limit ($\lambda_i \ll r_E$, $\lambda_{\text{post}} \ll r_A$), the prefactor $\lambda_{\text{post}} \lambda_i$ drives $B_i \to 0$ while the denominator remains $O(r_A^2 r_E^2)$. At higher rates, the drift becomes nonzero and rate-dependent, consistent with the simulation results in Condition 2. A full analysis of how this drift shifts the mean-field fixed point is deferred to future work; the numerical verification below treats the effect empirically.

## 5 Numerical Verification

### 5.1 Exact Event-Driven Update

The simulations below use an exact analytical integration of Equation 5 over each inter-event interval, avoiding numerical ODE solvers. Between consecutive events separated by an interval $\Delta$, the traces decay as $x_i(t) = x_{i0}\, e^{-r_E(t-t_0)}$ and $y(t) = y_0\, e^{-r_A(t-t_0)}$, where $x_{i0}$ and $y_0$ are the trace values at the start of the interval. Substituting into Equation 5 and integrating gives the exact weight increment:

$$\Delta w_i = \eta \left[ \frac{y_0^2\, x_{i0}\left(1 - e^{-(2r_A + r_E)\Delta}\right)}{2r_A + r_E} - \frac{y_0\, x_{i0}^2\left(1 - e^{-(r_A + 2r_E)\Delta}\right)}{r_A + 2r_E} \right]. \tag{41}$$

This expression is exact and reduces to the isolated-pair window (Equations 11–13) when one trace is at its reset value and $\Delta \to \infty$.

The event-driven simulation loop processes events in chronological order as follows:

1. **Integrate weights.** Compute the time $\Delta$ since the last event. Apply Equation 41 to all synapses using the current trace values $(x_{i0}, y_0)$.

2. **Decay traces.** Advance all trace variables to the current event time: $x_i \leftarrow x_{i0}\, e^{-r_E\Delta}$, $y \leftarrow y_0\, e^{-r_A\Delta}$.

3. **Reset trace.** If the event is a presynaptic spike from input $i$, set $x_i \leftarrow 1$. If the event is a postsynaptic spike, set $y \leftarrow 1$.

4. **Normalize.** If the event is a postsynaptic spike, apply multiplicative normalization: $w_k \leftarrow w_k / \sum_j w_j$ for all $k$.

5. **Record.** Log weight snapshots at regular intervals for analysis.

This chronology ensures that the weight update for each interval uses trace values from the *beginning* of the interval, before any reset or normalization occurs. After each weight update, weights are clipped to nonnegative values before normalization. In a representative sparse-regime run (Condition 1, seed 1000), clipping was active in approximately 1.3% of weight-update steps; the minimum final weight was $w_{\min} \approx 6.9 \times 10^{-6}$, well above zero. The clipping events are concentrated at low-target inputs where stochastic depression occasionally pushes a weight marginally below zero before the next normalization step. The discrete $+K$-then-normalize derivation in Section 4.3 should be read as a first-order mean-field abstraction of this event-driven loop. In the exact simulation, continuous kernel contributions are integrated over inter-event intervals and normalization is applied at postsynaptic events; differences in ordering are $O(K^2)$ under the small-$K$ approximation and are empirically absorbed into the residual drift term $B_i$.

### 5.2 Simulation Setup

We test the mean-field fixed-point prediction (Equation 35) with a reproducible, event-driven simulation. Seven conditions are reported: core convergence in the sparse regime, degradation across firing-rate regimes, comparison with classical trace-based STDP under identical normalization (with progressively matched parameters), hard-reset versus additive trace variants, a decorrelated control, an iso-rate control, and a heterogeneous-delay robustness check. All simulations use the exact inter-event weight update in Equation 41 and are run across 20 independent random seeds for statistical reporting. Code will be released in a public repository upon acceptance.

A single postsynaptic neuron receives $N = 100$ presynaptic inputs. Presynaptic spike trains are independent Poisson processes with heterogeneous rates $\nu_i$ drawn from a log-uniform distribution on $[1, 10]$ Hz (seed 42). Postsynaptic spikes are generated *open-loop*: each presynaptic spike from input $i$ independently triggers a postsynaptic spike with probability $q_i$, after a fixed causal delay of 5.0 ms. The triggering probabilities are set proportional to firing rate, $q_i = (\nu_i/\nu_{\max}) \times 0.10$, so that higher-rate inputs are also more likely to be causally linked to postsynaptic events. Open-loop generation means that the weights $w_i(t)$ do not influence

postsynaptic firing; this is appropriate for verifying the mean-field prediction, which assumes stationary statistics.

Weights are initialized uniformly at $w_i(0) = 1/N = 0.01$ and updated via the exact event-driven rule in Equation 41 with $\eta = 10^{-3}$, $r_E = 0.1782$ ms$^{-1}$, and $r_A = 0.0775$ ms$^{-1}$. At each postsynaptic spike, the weight vector is multiplicatively renormalized so that $\sum_k w_k = 1$. The theoretical target is $w_i^* = R_i/R_{\text{tot}}$ where $R_i = \nu_i q_i$ and $R_{\text{tot}} = \sum_j \nu_j q_j \approx 19.7$ Hz. Convergence is measured by the mean squared error $\text{MSE}(t) = N^{-1} \sum_i (w_i(t) - w_i^*)^2$ and by the Pearson correlation between $\mathbf{w}(t)$ and $\mathbf{w}^*$.

Each simulation runs for 600 s of simulated time. The full parameter specification is given in Table 2.

Table 2: Simulation parameters for the numerical verification.

| Parameter | Symbol | Value | Notes |
|---|---|---|---|
| Pre decay rate | $r_E$ | 0.1782 ms$^{-1}$ | Fitted (Section 3.1) |
| Post decay rate | $r_A$ | 0.0775 ms$^{-1}$ | Fitted (Section 3.1) |
| Learning rate | $\eta$ | $10^{-3}$ | Equation 5 |
| Number of inputs | $N$ | 100 | |
| Pre rates | $\nu_i$ | LogU(1, 10) Hz | Seed 42 |
| Post delay | — | 5.0 ms | Fixed causal delay |
| Post probability | $q_i$ | $(\nu_i/\nu_{\max}) \times 0.10$ | Open-loop |
| Normalization | — | After every post event | Multiplicative (Eq. 20) |
| Trace mode | — | Hard reset | $x_i(t^+) = 1$, $y(t^+) = 1$ |
| Duration | — | 600 s | Per seed |
| Seeds | — | 20 per condition | |

## 5.3  Condition 1: Core Fixed-Point Convergence

Figure 2a shows the MSE between the weight vector and the theoretical target across 20 seeds in the sparse regime ($\nu_i \in [1, 10]$ Hz). The MSE drops rapidly from its initial value of approximately $1.5 \times 10^{-4}$ and plateaus near $2.6 \times 10^{-5} \pm 2.8 \times 10^{-6}$ (mean $\pm$ std across seeds) by $t = 600$ s. Figure 2b shows individual weight trajectories from a representative seed, converging toward their respective targets (dashed lines). Figure 2c plots final weights against theoretical targets across all 100 inputs, averaged over seeds. The Pearson correlation is $r = 0.918 \pm 0.014$.

The scatter plot shows a systematic compression of the highest-target weights, consistent with a small residual background drift $B_i$ (Equation 19) that is not exactly zero even in the sparse regime. This effect is explored further in Condition 2.

## 5.4  Condition 2: Sparse versus Dense Firing Regimes

To quantify the background drift $B_i$, the simulation was repeated across four firing-rate regimes: very sparse (0.5-2 Hz), sparse (2-10 Hz), moderate (10-50 Hz), and dense (50-200 Hz). All other parameters were held constant except duration, which was scaled inversely with mean rate to keep the total number of postsynaptic events roughly comparable. Figure 3a shows that the final MSE increases monotonically across regimes: $8.1 \times 10^{-6}$ (very sparse), $1.9 \times 10^{-5}$ (sparse), $3.0 \times 10^{-5}$ (moderate), and $3.8 \times 10^{-5}$ (dense), a 4.6-fold degradation from the sparsest to densest condition.

This progression is consistent with the theoretical expectation. The isolated-pair balance theorem (Equation 18) holds exactly only for non-overlapping spike pairs. At higher rates, trace overlap introduces nonzero $B_i$ through higher-order interactions, systematically shifting the weights away from the clean mean-field target. The very sparse regime, where inter-spike intervals greatly exceed trace timescales ($1/r_A \approx 12.9$ ms, $1/r_E \approx 5.6$ ms), achieves the closest match to the predicted fixed point.

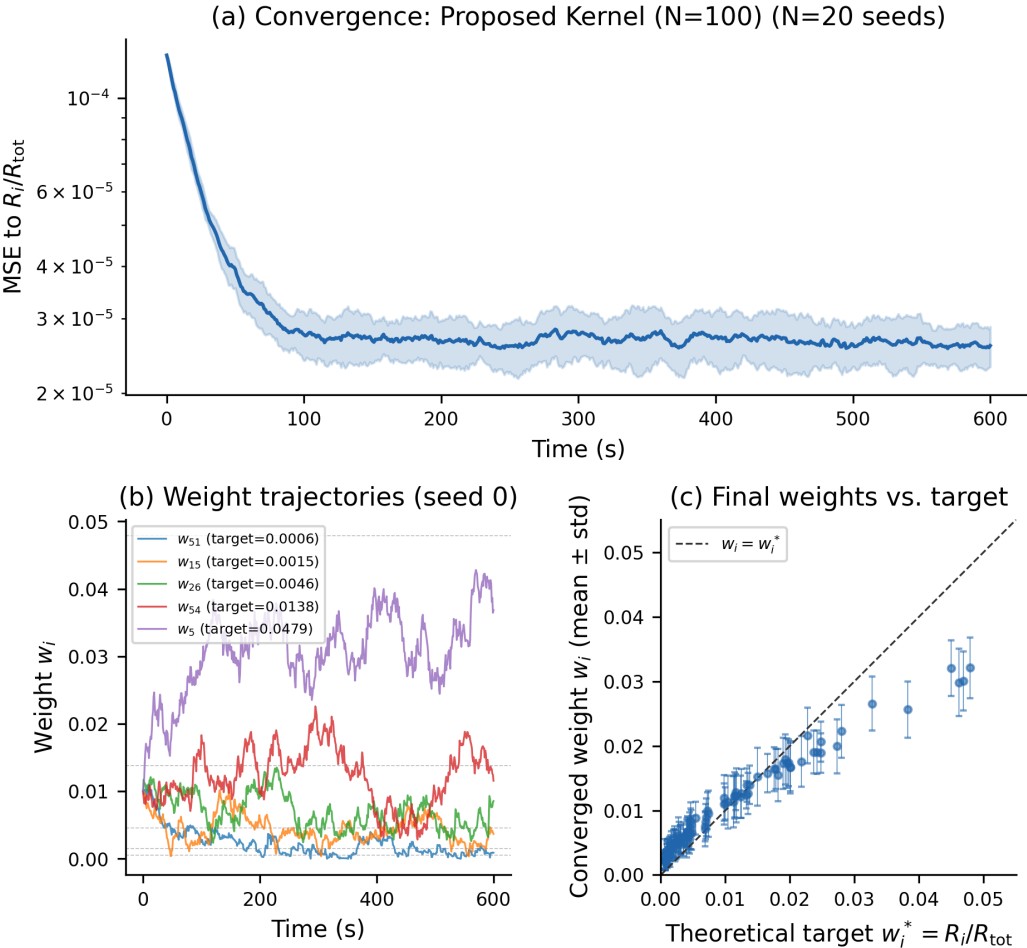

Figure 2: Core convergence (Condition 1). **(a)** MSE to the theoretical target $w_i^* = R_i/R_{\text{tot}}$ across 20 seeds (solid line: mean; shading: $\pm 1$ std). **(b)** Weight trajectories for five representative inputs from seed 1000, with dashed lines indicating theoretical targets. **(c)** Final weights (mean $\pm$ std across seeds) versus theoretical target for all 100 inputs. The dashed diagonal is $w_i = w_i^*$.

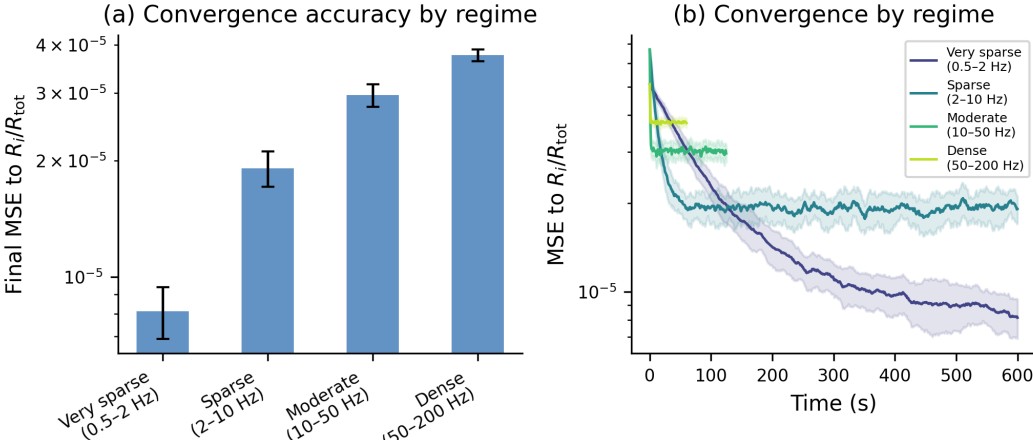

Figure 3: Convergence accuracy across firing-rate regimes (Condition 2). **(a)** Final MSE to target (mean $\pm$ std, 20 seeds) for four regimes. **(b)** MSE trajectories over time. Very sparse inputs converge most accurately; dense inputs plateau at higher residual error, consistent with nonzero background drift $B_i$.

### 5.5 Condition 3: Classical STDP Baselines

To test whether convergence to the attribution target is specific to the proposed kernel or is merely a byproduct of multiplicative normalization, we compared against classical trace-based pairwise STDP under identical conditions. Five classical baselines were evaluated, progressively controlling for area balance, trace mode, effective step size, and temporal footprint:

- **Unbalanced, additive traces** ($A_+ = 0.005$, $A_- = 0.00525$, $\tau_+ = \tau_- = 20$ ms): the standard LTD-dominant parameterization (Song et al., 2000), with net-negative integrated area.

- **Area-balanced, additive traces** ($A_+ = A_- = 0.005$, $\tau_+ = \tau_- = 20$ ms): controls for area imbalance alone.

- **Area-balanced, hard-reset traces** (same amplitudes and time constants, hard-reset): additionally controls for trace mode.

- **Matched-$K$, hard-reset** ($A_+ = A_- = 0.001083$, $\tau_+ = \tau_- = 20$ ms, hard-reset): the LTP amplitude is set so that the effective step at the causal delay matches the proposed kernel, $A_+ \exp(-5/\tau_+) = K_{\text{proposed}}$. This controls for step-size differences.

- **Matched-$K$ and temporal footprint, hard-reset** ($A_+ = A_- = 0.001518$, $\tau_+ = \tau_- = 8.5$ ms, hard-reset): the time constant is set to the geometric mean of the proposed kernel's characteristic timescales ($\sqrt{r_E^{-1} r_A^{-1}} \approx 8.5$ ms), with $A_+$ adjusted to preserve the matched step size. This controls for both step size and temporal window width.

Figure 4 shows that the proposed kernel (MSE $= 2.6 \times 10^{-5}$, Pearson $r = 0.918$) outperforms all five classical baselines. Moving up the control ladder: the unbalanced additive baseline (MSE $= 1.0 \times 10^{-4}$) performs worst. Balancing the area with additive traces improves to MSE $= 9.5 \times 10^{-5}$. Switching to hard-reset traces improves further (MSE $= 8.4 \times 10^{-5}$). Matching the effective step size brings a substantial improvement (MSE $= 4.6 \times 10^{-5}$), and matching the temporal footprint brings the classical rule closest (MSE $= 3.5 \times 10^{-5}$). Even in this most controlled comparison, the proposed kernel retains a $1.3\times$ MSE advantage.

This decomposition controls four major confounds: area balance, trace mode, effective step size, and temporal footprint. The remaining gap between the best-matched classical baseline and the proposed kernel suggests an additional contribution from the smooth biphasic shape of the cubic window, which concentrates its positive lobe more precisely around the causal peak than the symmetric exponential decay of classical STDP.

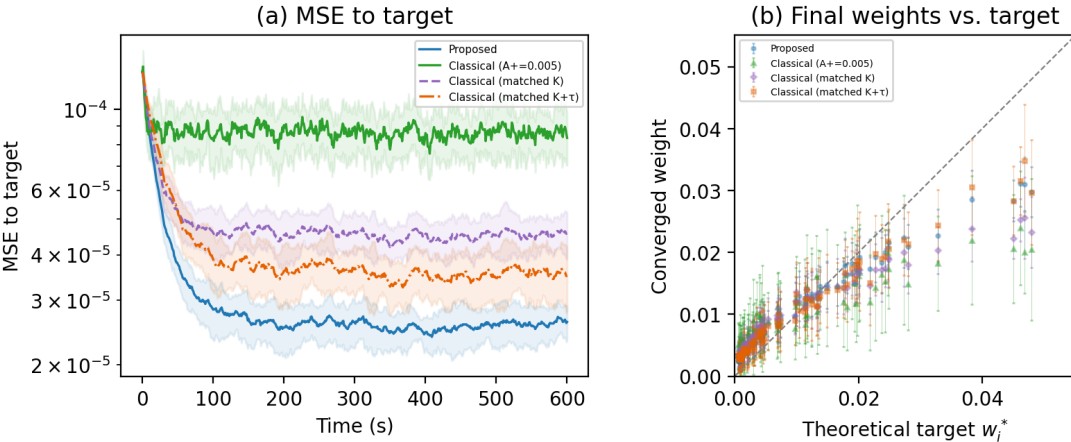

Figure 4: Baseline comparison (Condition 3). **(a)** MSE to target for the proposed kernel and five classical STDP baselines: unbalanced additive, area-balanced additive, area-balanced hard-reset, matched-$K$ hard-reset, and matched-$K$-and-$\tau$ hard-reset, under identical normalization and event streams. The area-balanced additive baseline is omitted from the plot for visual clarity but reported in Table 3. **(b)** Final weights versus theoretical target. The proposed kernel (blue circles) tracks the diagonal most closely; progressively matched classical baselines approach but do not reach the proposed kernel's accuracy.

### 5.6 Condition 4: Hard-Reset versus Additive Traces

The mean-field analysis assumes nearest-spike hard-reset traces (Equation 4), in which each new spike sets the trace to exactly 1 regardless of its prior value. An alternative is additive traces, in which each spike increments the trace by 1 ($x_i(t^+) = x_i(t)+1$), allowing amplitude accumulation from rapid successive spikes. We compared both variants on the same sparse network used in Condition 1.

Hard-reset traces achieved a final MSE of $2.6 \times 10^{-5}$; additive traces achieved $4.9 \times 10^{-5}$, roughly $1.8\times$ worse (Figure 5). The degradation under additive traces reflects the breakdown of the Markovian nearest-spike assumption: when traces accumulate amplitude, the effective kernel deviates from the isolated-pair window analyzed in Section 3, introducing additional background drift. This result supports the modeling choice of hard-reset traces for the regime where the mean-field analysis is intended to apply. The additive-trace variant of the proposed kernel remains better than the unbalanced and unmatched classical baselines, but it no longer outperforms the best matched classical baseline from Condition 3 (MSE = $3.5 \times 10^{-5}$). The cleanest evidence for the cubic kernel's residual advantage is therefore the hard-reset proposed kernel (MSE = $2.6 \times 10^{-5}$) compared against the matched-$K$, matched-$\tau$, hard-reset classical baseline (MSE = $3.5 \times 10^{-5}$).

### 5.7 Condition 5: Decorrelated Attribution Control

In Conditions 1–4, the triggering probabilities are set proportional to firing rate ($q_i = (\nu_i/\nu_{\max}) \times 0.10$), so the target weight $w_i^* \propto \nu_i q_i \propto \nu_i^2$. This raises the concern that the network may simply amplify high-rate inputs rather than track independent attribution probabilities. To address this, we repeated the sparse-regime simulation with $q_i$ drawn independently of $\nu_i$ from a uniform distribution on $[0.01, 0.10]$, yielding a near-zero input correlation ($r(\nu_i, q_i) = 0.009$).

The proposed kernel converges with MSE = $1.8 \times 10^{-5}$ and per-seed Pearson $r = 0.839 \pm 0.023$ (mean $\pm$ std across 20 seeds) to the theoretical target $w_i^* = \nu_i q_i / \sum_j \nu_j q_j$ (Figure 6a–b). To test whether the network tracks the *product* $\nu_i q_i$ and not just firing rate, we computed the correlation between the seed-averaged final weights and three candidate predictors: $\nu_i q_i$ (the target, $r = 0.934$), $\nu_i$ alone ($r = 0.924$), and $q_i$ alone ($r = 0.310$). The high correlation with $\nu_i$ alone reflects the fact that both the number of eligible events and the target scale with rate. However, a partial correlation analysis controlling for $\nu_i$ yields $r(w, q_i \mid \nu_i) = 0.79$,

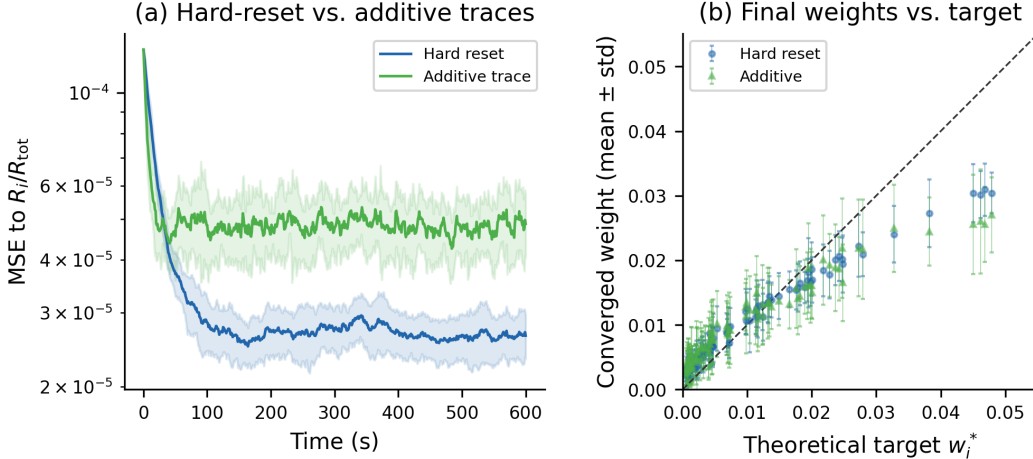

Figure 5: Trace variant comparison (Condition 4). **(a)** MSE to target for hard-reset traces (blue) and additive traces (green). **(b)** Final weights versus target. Both variants track the target qualitatively, but hard-reset traces are more accurate, consistent with the Markovian assumption in the theory.

confirming that the weights encode substantial attribution-probability information beyond raw firing rate, even though firing rate remains a strong predictor because eligible event counts scale with $\nu_i$.

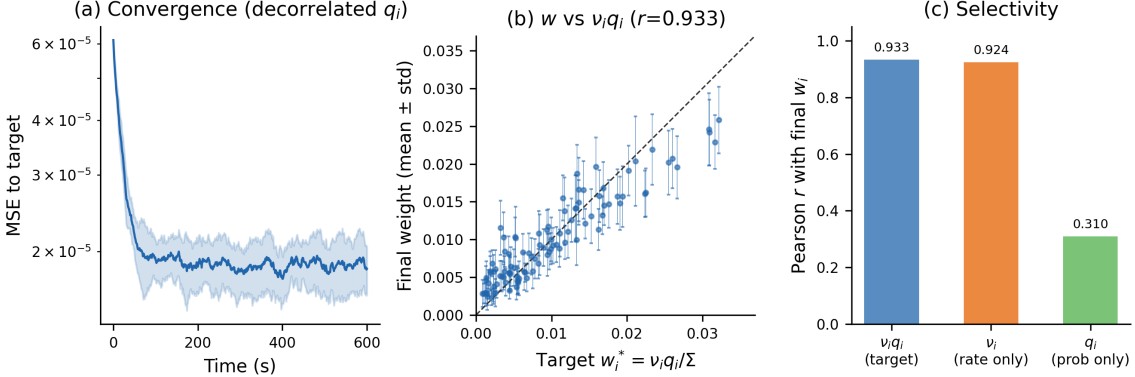

Figure 6: Decorrelated attribution control (Condition 5). $q_i$ drawn independently of $\nu_i$. **(a)** MSE to target over time. **(b)** Final weights vs. theoretical target $w_i^* = \nu_i q_i/\Sigma$. **(c)** Pearson correlation of mean final weights with the target ($\nu_i q_i$), firing rate alone ($\nu_i$), and triggering probability alone ($q_i$).

### 5.8 Condition 6: Iso-Rate Attribution Control

The decorrelated control in Condition 5 reduces the $q_i$–$\nu_i$ correlation but does not eliminate rate as a confound, because both the number of eligible events and the target scale with $\nu_i$. To test whether the network can track attribution probability independent of firing rate, we set all presynaptic rates to a constant $\nu_i = 5$ Hz and drew $q_i \sim \text{Uniform}(0.01, 0.10)$ independently. The target reduces to $w_i^* \propto q_i$.

The proposed kernel converges with MSE $= 2.1 \times 10^{-5}$. Individual seeds show moderate per-seed correlations ($r = 0.426 \pm 0.060$) due to stochastic noise at equal rates, but the seed-averaged weight vector correlates strongly with $q_i$ ($r(\mathbf{w}_{\text{mean}}, q_i) = 0.903$; Figure 7). Because all inputs fire at the same rate, any weight differentiation must arise from the attribution probabilities alone. The high seed-averaged correlation confirms that the network reliably separates attribution probability from firing rate.

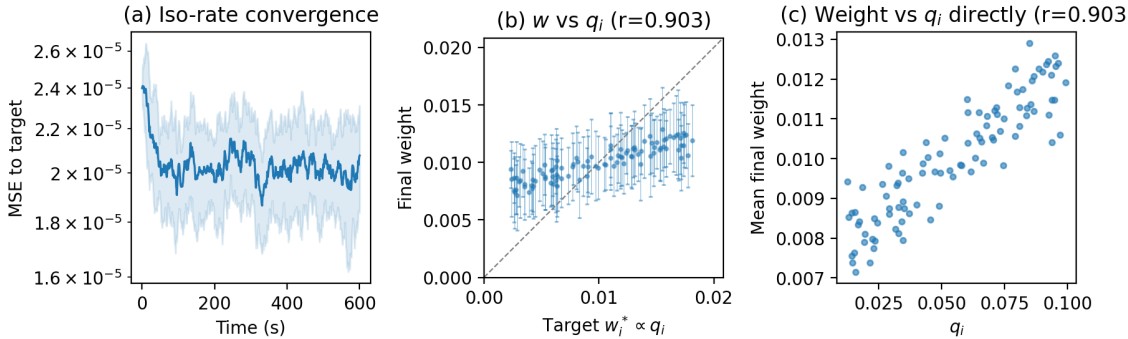

Figure 7: Iso-rate attribution control (Condition 6). All $\nu_i = 5$ Hz; $q_i \sim U(0.01, 0.10)$. **(a)** MSE over time. **(b)** Final weights vs. target $w_i^* \propto q_i$. **(c)** Weight vs. $q_i$ directly, showing clear monotonic tracking.

### 5.9 Condition 7: Heterogeneous Causal Delays

All preceding conditions use a fixed causal delay of 5.0 ms. To test robustness to heterogeneous timing, we drew per-input delays $d_i \sim \text{Uniform}(2, 10)$ ms and compared convergence against both the kernel-weighted target $w_i^* \propto \nu_i q_i K_i$ (where $K_i = \eta W(d_i)$ depends on the input-specific delay) and the simpler target $w_i^* \propto \nu_i q_i$.

The network converges with MSE $= 2.8 \times 10^{-5}$ and Pearson $r = 0.912$ to the kernel-weighted target, slightly better than its match to the simple target ($r = 0.899$). This confirms that the learned fixed point reflects the full kernel-weighted attribution statistic, not just the product $\nu_i q_i$, and that the rule is not overfit to a single delay.

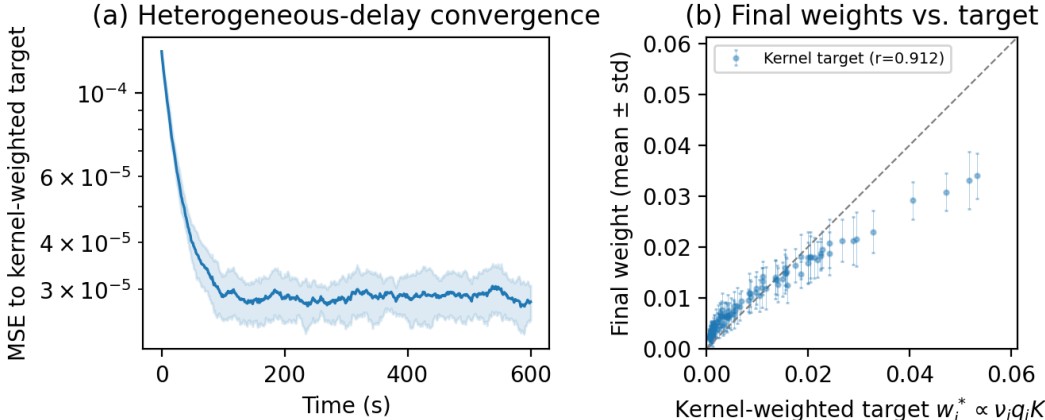

Figure 8: Heterogeneous-delay robustness (Condition 7). Delays $d_i \sim U(2, 10)$ ms. **(a)** MSE to the kernel-weighted target over time. **(b)** Final weights vs. kernel-weighted target $w_i^* \propto \nu_i q_i K_i$.

### 5.10 Summary of Numerical Results

Table 3 summarizes the key quantitative outcomes across conditions.

Table 3: Summary of numerical verification results (20 seeds per condition).

| Condition | Final MSE (mean $\pm$ std) | Pearson $r$ | Key finding |
|---|---|---|---|
| 1. Core convergence | $2.59\mathrm{e}{-5} \pm 2.84\mathrm{e}{-6}$ | $0.918 \pm 0.014$ | Weights track $w_i^*$ |
| 2a. Very sparse (0.5–2 Hz) | $8.15\mathrm{e}{-6} \pm 1.25\mathrm{e}{-6}$ | $0.921 \pm 0.013$ | $B_i \approx 0$ |
| 2b. Sparse (2–10 Hz) | $1.91\mathrm{e}{-5} \pm 2.03\mathrm{e}{-6}$ | $0.866 \pm 0.021$ | — |
| 2c. Moderate (10–50 Hz) | $2.96\mathrm{e}{-5} \pm 1.99\mathrm{e}{-6}$ | $0.835 \pm 0.024$ | — |
| 2d. Dense (50–200 Hz) | $3.76\mathrm{e}{-5} \pm 1.32\mathrm{e}{-6}$ | $0.650 \pm 0.033$ | $B_i$ non-negligible |
| 3a. Classical unbal. (additive) | $1.01\mathrm{e}{-4} \pm 1.62\mathrm{e}{-5}$ | $0.519$ | Area imbalance hurts |
| 3b. Classical bal. (additive) | $9.48\mathrm{e}{-5} \pm 1.54\mathrm{e}{-5}$ | $0.550$ | Balance helps |
| 3c. Classical bal. (hard-reset) | $8.35\mathrm{e}{-5} \pm 1.33\mathrm{e}{-5}$ | $0.592$ | Reset helps further |
| 3d. Classical matched-$K$ (hard-reset) | $4.57\mathrm{e}{-5} \pm 6.46\mathrm{e}{-6}$ | $0.834$ | Step-size matching helps |
| 3e. Classical matched-$K+\tau$ | $3.51\mathrm{e}{-5} \pm 7.01\mathrm{e}{-6}$ | $0.856$ | Cubic form still adds |
| 4. Additive traces | $4.87\mathrm{e}{-5} \pm 8.33\mathrm{e}{-6}$ | — | Hard reset preferred |
| 5. Decorrelated $q_i$ | $1.82\mathrm{e}{-5} \pm 2.29\mathrm{e}{-6}$ | $0.839$ | Tracks $\nu_i q_i$ |
| 6. Iso-rate ($\nu_i = 5$ Hz) | $2.08\mathrm{e}{-5} \pm 2.08\mathrm{e}{-6}$ | $0.426^\dagger$ | Tracks $q_i$ alone |
| 7. Heterogeneous delays | $2.78\mathrm{e}{-5} \pm 2.94\mathrm{e}{-6}$ | $0.912$ | Tracks kernel-weighted target |

$^\dagger$Per-seed $r$ is low due to stochastic noise at equal rates; seed-averaged $r(\mathbf{w}_{\mathrm{mean}}, q_i) = 0.903$.

The simulations confirm five predictions. First, in the sparse regime, the proposed kernel with multiplicative normalization converges to the theoretical fixed point $w_i^* = \nu_i q_i / \sum_j \nu_j q_j$ with high accuracy. Second, the background drift $B_i$ is approximately zero when inter-spike intervals are large relative to trace timescales, and becomes progressively non-negligible at higher rates, exactly as anticipated in Section 3.5 and consistent with the analytic expression in Equation 40. Third, the convergence to the normalized attribution target is not an artifact of normalization alone: even with area balance, matched trace mode, matched step size, and matched temporal footprint, classical STDP does not reach the proposed kernel's accuracy. Fourth, the network tracks attribution probability independently of firing rate, as confirmed by both the decorrelated control (Condition 5) and the iso-rate control (Condition 6). Fifth, the rule is robust to heterogeneous causal delays (Condition 7), with weights tracking the kernel-weighted attribution target.

## 6 Discussion

### 6.1 Overview

We presented a continuous trace-interaction kernel for STDP and derived its isolated-pair learning window in closed form. The main mathematical result is that the total integrated area of the isolated-pair window is exactly zero for all positive decay rates. This gives the kernel an intrinsic balance property without tuning separate LTP and LTD amplitudes.

The fitted window also contains a narrow near-coincident positive-update regime for small post-before-pre delays. This is a structural consequence of the fitted continuous kernel ($r_E > r_A$), but the raw data in the corresponding interval are consistent with it. Forcing the zero crossing to $\Delta t = 0$ substantially worsens the fit. This does not undermine the exact balance theorem, but it does narrow the interpretation of the mean-field result. The strict posterior-attribution claim holds under a causal-window approximation in which positive evidence is identified with pre-before-post timing. The full fitted kernel should be interpreted more generally as learning a normalized kernel-weighted attribution statistic that may include causal and near-coincident temporal relationships.

At the network level, the strongest result is Equation 35: with afferent normalization, additive positive attribution events produce a delta-rule-like convergence to normalized event-rate targets. Under additional assumptions, this target becomes $P(\mathrm{Pre}_i \mid \mathrm{Post})$. Without those assumptions, it is more precise to describe the fixed point as normalized conditional event-rate attribution rather than as a literal posterior probability. The convergence advantage of the proposed kernel over classical STDP persists even when the classical baseline is given area balance, hard-reset traces, matched step size, and matched temporal footprint ($1.3\times$

MSE advantage in the most controlled comparison). An iso-rate control and a decorrelated control further confirm that the network tracks attribution probability, not firing rate alone, and a heterogeneous-delay condition confirms robustness to non-uniform timing.

## 6.2 Limitations

Several limitations are important. First, the traces analyzed here are nearest-spike hard-reset variables. This is a useful Markovian abstraction, but it does not capture all burst, triplet, calcium-accumulation, or rate-dependent effects associated with biological STDP. Second, the exact zero-area theorem applies to isolated spike pairs integrated over the full decay tails. Dense spike trains, overlapping traces, and hard resets can introduce nonzero background drift, represented by $B_i$ in Equation 19. Third, multiplicative afferent normalization is a modeling abstraction for postsynaptic competition and homeostasis. The normalization constraint $\sum_k w_k = 1$ is local to each postsynaptic neuron's afferent weight vector (analogous to synaptic scaling or dendritic homeostasis), not a network-wide global operation. However, even neuron-local normalization requires an $O(N)$ summation and division over the fan-in, which is not strictly synapse-local. In biological circuits, homeostatic synaptic scaling operates on slower timescales than individual plasticity events (Turrigiano, 2017). In neuromorphic hardware, enforcing exact normalization after every postsynaptic spike would require a synchronous read-sum-divide operation across all afferent synapses, imposing communication and latency costs. The instantaneous normalization used here should therefore be understood as an idealized fixed-point mechanism; relaxing it to periodic or approximate normalization is an important direction for deployment-oriented extensions.

Fourth, the probabilistic interpretation depends on the normalization axis and on event-statistical assumptions. The step from normalized event-rate attribution to the posterior $P(\text{Pre}_i \mid \text{Post})$ (Equation 37) requires that presynaptic events form a mutually exclusive and exhaustive partition for the postsynaptic event. The 100 independent Poisson inputs in the simulation can fire simultaneously, technically violating this assumption. In the sparse regime, the probability of simultaneous triggering is negligible (each $q_i \leq 0.10$), so the partition approximation is practically accurate. In denser regimes, this approximation would degrade. Afferent normalization produces relative attribution among inputs to a postsynaptic unit; it does not generally recover raw $P(\text{Post} \mid \text{Pre}_i)$ values. Finally, this paper does not claim task-level machine-learning performance. Functional benchmarks, wall-clock profiling, and comparisons with established STDP variants are future empirical work, not evidence supplied by the current theory.

## 6.3 Future Work

Immediate future work is to study the near-coincident positive-update regime as a broader temporal-association signal rather than as a strict causal window, and to investigate whether the continuous cubic kernel's advantage over area-balanced classical STDP arises from its smooth biphasic window shape, its interaction with the hard-reset trace dynamics, or both. A second direction is to embed the rule in larger spiking networks and closed-loop task environments once the theory-level numerical checks are complete.

### Broader Impact Statement

This work is primarily theoretical, developing a mathematical framework connecting spike-timing-dependent plasticity to probabilistic credit assignment. We do not foresee direct negative societal impacts from this contribution. The framework could eventually inform neuromorphic hardware design or biologically inspired learning algorithms, but the present paper reports no deployable system or application.

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

## A  Parameter Sensitivity of the Isolated-Pair Window

This appendix summarizes the finite-window parameter sweep used to characterize the isolated-pair STDP window. The sweep evaluates all 16 combinations of

$$r_A, r_E \in \{0.05, 0.1, 0.2, 0.4\}\,\text{ms}^{-1},$$

where $r_A$ is the postsynaptic trace decay rate and $r_E$ is the presynaptic trace decay rate. For each pair, the closed-form window in Equations 11–13 was sampled over a finite temporal range and summarized by the metrics in Table 4.

Figure 9: Isolated-pair STDP windows across the decay-parameter sweep, with $r_A, r_E \in \{0.05, 0.1, 0.2, 0.4\}\,\mathrm{ms}^{-1}$. Each panel plots the closed-form window $\Delta w(\Delta t)$ from Equations 11–13 over $\Delta t \in [-100, 100]\,\mathrm{ms}$. Red dotted lines mark the primary zero crossing (ZC) separating the main potentiation and depression lobes, with the interpolated value noted in each panel. When $r_A = r_E$ (panels a, f, k, p), the window is antisymmetric and the zero crossing falls at $\Delta t \approx 0$. When $r_E > r_A$ (below the diagonal), the zero crossing shifts to negative $\Delta t$, producing a narrow near-coincident positive-update regime for small post-before-pre lags; when $r_A > r_E$ (above the diagonal), the crossing shifts to positive $\Delta t$. Increasing $r_A$ reduces peak potentiation; increasing $r_E$ reduces peak depression. These trends are consistent with the finite-window metrics in Table 4.

The metrics are defined as follows. $\mathrm{LTP}_{\max} = \max_{\Delta t} \Delta w(\Delta t)$, $\Delta t_{\mathrm{LTP}} = \arg\max_{\Delta t} \Delta w(\Delta t)$, $\mathrm{LTD}_{\max} = \min_{\Delta t} \Delta w(\Delta t)$, and $\Delta t_{\mathrm{LTD}} = \arg\min_{\Delta t} \Delta w(\Delta t)$. The dynamic range is $\Delta w_{\mathrm{range}} = \mathrm{LTP}_{\max} - \mathrm{LTD}_{\max}$, the peak ratio is $R_{\mathrm{peak}} = \mathrm{LTP}_{\max}/|\mathrm{LTD}_{\max}|$, and the peak-to-peak separation is $\Delta t_{\mathrm{pp}} = \Delta t_{\mathrm{LTP}} - \Delta t_{\mathrm{LTD}}$. The finite-window areas are

$$A_{\mathrm{LTP}} = \int_{\{\Delta t : \Delta w(\Delta t) > 0\}} \Delta w(\Delta t)\, d\Delta t, \qquad A_{\mathrm{LTD}} = -\int_{\{\Delta t : \Delta w(\Delta t) < 0\}} \Delta w(\Delta t)\, d\Delta t,$$

and $R_{\mathrm{area}} = A_{\mathrm{LTP}}/A_{\mathrm{LTD}}$. The zero crossing $\Delta t_{\mathrm{ZC}}$ denotes the primary crossing separating the main depression and potentiation lobes.

Table 4: Finite-window metrics across the decay-parameter sweep. These values numerically summarize the isolated-pair window; they are not a substitute for the exact balance proof in Equation 18.

| Case | $r_A$ | $r_E$ | $\text{LTP}_{\max}$ | $\Delta t_{\text{LTP}}$ | $\text{LTD}_{\max}$ | $\Delta t_{\text{LTD}}$ | Range | $R_{\text{peak}}$ | $\Delta t_{\text{pp}}$ | $A_{\text{LTP}}$ | $A_{\text{LTD}}$ | $R_{\text{area}}$ | $\Delta t_{\text{ZC}}$ |
|------|-------|-------|------|------|------|------|-------|-------|------|------|------|------|------|
| (a) | 0.05 | 0.05 | 1.668 | 13.86 | -1.668 | -13.86 | 3.335 | 1.000 | 27.72 | 65.77 | 65.79 | 0.9997 | -0.01 |
| (b) | 0.10 | 0.05 | 0.801 | 18.32 | -1.563 | -4.70 | 2.364 | 0.512 | 23.02 | 31.49 | 32.00 | 0.9841 | 4.45 |
| (c) | 0.20 | 0.05 | 0.371 | 21.96 | -1.250 | -1.44 | 1.621 | 0.297 | 23.40 | 14.54 | 14.80 | 0.9824 | 8.09 |
| (d) | 0.40 | 0.05 | 0.174 | 24.44 | -0.849 | -0.41 | 1.023 | 0.204 | 24.85 | 6.79 | 6.90 | 0.9835 | 10.57 |
| (e) | 0.05 | 0.10 | 1.563 | 4.70 | -0.801 | -18.32 | 2.364 | 1.952 | 23.02 | 32.00 | 31.49 | 1.0162 | -4.46 |
| (f) | 0.10 | 0.10 | 0.834 | 6.93 | -0.834 | -6.93 | 1.668 | 1.000 | 13.86 | 16.67 | 16.67 | 1.0000 | -0.01 |
| (g) | 0.20 | 0.10 | 0.401 | 9.15 | -0.782 | -2.35 | 1.183 | 0.512 | 11.50 | 8.01 | 8.00 | 1.0015 | 2.22 |
| (h) | 0.40 | 0.10 | 0.186 | 10.97 | -0.625 | -0.72 | 0.811 | 0.297 | 11.69 | 3.72 | 3.70 | 1.0049 | 4.04 |
| (i) | 0.05 | 0.20 | 1.250 | 1.44 | -0.371 | -21.96 | 1.621 | 3.369 | 23.40 | 14.80 | 14.54 | 1.0179 | -8.10 |
| (j) | 0.10 | 0.20 | 0.782 | 2.35 | -0.401 | -9.15 | 1.183 | 1.951 | 11.50 | 8.00 | 8.01 | 0.9986 | -2.23 |
| (k) | 0.20 | 0.20 | 0.418 | 3.46 | -0.418 | -3.46 | 0.835 | 1.000 | 6.92 | 4.17 | 4.17 | 1.0000 | -0.01 |
| (l) | 0.40 | 0.20 | 0.201 | 4.57 | -0.391 | -1.17 | 0.592 | 0.513 | 5.74 | 2.01 | 2.00 | 1.0032 | 1.11 |
| (m) | 0.05 | 0.40 | 0.849 | 0.41 | -0.174 | -24.44 | 1.023 | 4.891 | 24.85 | 6.90 | 6.79 | 1.0168 | -10.58 |
| (n) | 0.10 | 0.40 | 0.625 | 0.72 | -0.186 | -10.97 | 0.811 | 3.363 | 11.69 | 3.70 | 3.72 | 0.9951 | -4.05 |
| (o) | 0.20 | 0.40 | 0.391 | 1.17 | -0.201 | -4.57 | 0.592 | 1.949 | 5.74 | 2.00 | 2.01 | 0.9969 | -1.12 |
| (p) | 0.40 | 0.40 | 0.209 | 1.73 | -0.209 | -1.73 | 0.418 | 1.000 | 3.46 | 1.04 | 1.04 | 1.0000 | -0.01 |
| Fitted | 0.0775 | 0.1782 | 0.978 | 2.41 | -0.443 | -12.34 | 1.421 | 2.206 | 14.75 | 11.42 | 11.42 | 0.9995 | -3.41 |

The sweep illustrates three qualitative relationships. First, when $r_A = r_E$, the two lobes are approximately symmetric in timing and peak magnitude. Second, increasing $r_A$ shortens the postsynaptic trace and reduces peak potentiation. Third, increasing $r_E$ shortens the presynaptic trace and reduces peak depression. The numerical area ratios remain close to one across the grid, as expected from the exact theorem, with deviations reflecting finite-window sampling and numerical approximation.

## B  Alternative Empirical Fit Sensitivity

The main text reports the conservative raw-data fit to the classical measurements of Bi & Poo (1998). We also examined two sliding-window summaries of the same raw data: a local mean and a local max-absolute envelope. These fits are sensitivity analyses only. They show that the same kernel shape can match smoothed summaries of noisy empirical data, but they are not used as the primary evidence for the kernel and should not be interpreted as replacing the raw-data fit.

In both analyses, a sliding window of width $5\,\text{ms}$ was moved over the observed inter-spike intervals. For a window centered at $t_k$, the local-mean target was

$$\Delta w_{\text{mean}}(t_k) = \frac{1}{n_k} \sum_{(\Delta t_j, \Delta w_j) \in D_k} \Delta w_j,$$

where $D_k$ is the set of raw points in the window. The local-envelope target retained the signed value with largest absolute magnitude in the window,

$$\Delta w_{\text{env}}(t_k) = \Delta w_{j^*}, \qquad j^* = \arg \max_{j \in D_k} |\Delta w_j|.$$

The model was then refit to each summary using the same scale-factor approach as Equation 7.

Table 5: Empirical fit sensitivity across raw and smoothed target summaries. The raw-data fit remains the primary fit reported in the main text.

| Method | $r_A$ | $r_E$ | $c$ | $R^2$ | $\text{LTP}_{\max}$ | $\Delta t_{\text{LTP}}$ | $\text{LTD}_{\max}$ | $\Delta t_{\text{LTD}}$ | Range | $R_{\text{peak}}$ | $\Delta t_{\text{pp}}$ | $A_{\text{LTP}}$ | $A_{\text{LTD}}$ | $R_{\text{area}}$ | ZC |
|--------|-------|-------|-----|-------|------|------|------|------|-------|-------|------|------|------|------|------|
| Raw data | 0.0775 | 0.1782 | 106.5064 | 0.6340 | 0.978 | 2.41 | -0.443 | -12.34 | 1.421 | 2.206 | 14.75 | 11.420 | 11.420 | 0.9995 | -3.41 |
| Sliding-window mean | 0.0610 | 0.1200 | 47.16 | 0.8962 | 1.286 | 3.96 | -0.669 | -14.93 | 1.954 | 1.923 | 18.89 | 21.893 | 21.791 | 1.0050 | -3.58 |
| Sliding-window max-absolute | 0.0375 | 0.1200 | 61.72 | 0.8457 | 1.825 | 2.84 | -0.634 | -27.88 | 2.459 | 2.879 | 30.72 | 33.750 | 31.574 | 1.0690 | -9.40 |

The smoothed fits produce higher $R^2$ values than the raw fit, as expected when noisy point-to-point variation is summarized before fitting. They also preserve the key qualitative behavior of the fitted kernel, including a negative-lag zero crossing. Because smoothing choices can change the target being fit, these analyses are best interpreted as robustness checks on window morphology rather than as stronger evidence than the raw-data fit.

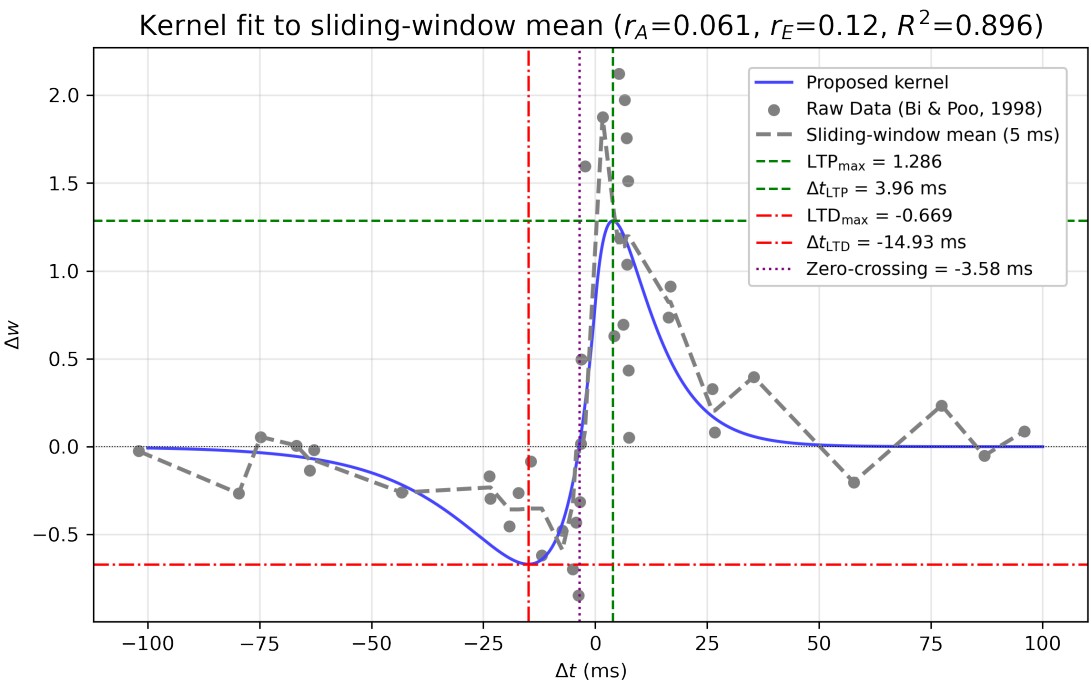

Figure 10: Kernel fit to the sliding-window local mean of the Bi and Poo (1998) data. Gray circles show the raw empirical measurements scaled by $1/c$; the black dashed curve shows the local mean computed in a 5 ms sliding window. The blue curve is the proposed kernel with fitted parameters $r_A = 0.0610\,\text{ms}^{-1}$, $r_E = 0.1200\,\text{ms}^{-1}$, $c = 47.16$ ($R^2 = 0.896$). Green and red lines mark the peak potentiation (LTP$_{\text{max}} = 1.286$ at $\Delta t = 3.96\,\text{ms}$) and peak depression (LTD$_{\text{max}} = -0.669$ at $\Delta t = -14.93\,\text{ms}$), respectively. The purple dotted line marks the primary zero crossing at $\Delta t = -3.58\,\text{ms}$. This fit is a sensitivity analysis of the same kernel shape applied to a smoothed summary of the raw data; it is not used as the primary evidence for the kernel and should not be interpreted as replacing the raw-data fit reported in Section 3.1.

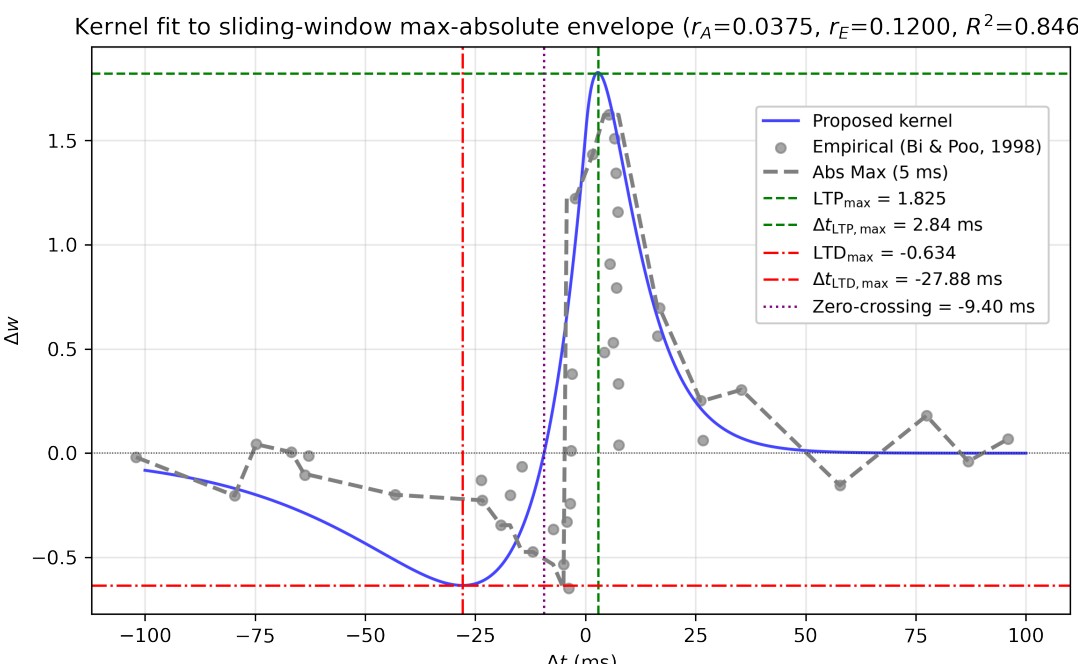

Figure 11: Kernel fit to the sliding-window max-absolute envelope of the Bi and Poo (1998) data. Gray circles show the raw empirical measurements scaled by $1/c$; the black dashed curve shows the signed value with largest absolute magnitude in each $5\,\mathrm{ms}$ window. The blue curve is the proposed kernel with fitted parameters $r_A = 0.0375\,\mathrm{ms}^{-1}$, $r_E = 0.1200\,\mathrm{ms}^{-1}$, $c = 61.72$ ($R^2 = 0.846$). Green and red lines mark peak potentiation ($\mathrm{LTP}_{\max} = 1.825$ at $\Delta t = 2.84\,\mathrm{ms}$) and peak depression ($\mathrm{LTD}_{\max} = -0.634$ at $\Delta t = -27.88\,\mathrm{ms}$), respectively. The purple dotted line marks the primary zero crossing at $\Delta t = -9.40\,\mathrm{ms}$. The wider LTD lobe and more negative zero crossing relative to the local-mean fit (Figure 10) reflect the envelope's sensitivity to extreme negative values at moderate lags. As with the local-mean fit, this analysis is a robustness check on window morphology rather than a replacement for the raw-data fit.

## C    Component Ablations

The full kernel can be decomposed into a multiplicative gating component and a competitive comparison component:

$$\frac{dw_i}{dt} = \eta \underbrace{y(t)x_i(t)}_{\text{overlap gate}} \underbrace{\big(y(t) - x_i(t)\big)}_{\text{trace comparator}}.$$

The ablations below show that neither component alone produces the full biphasic, timing-sensitive learning window.

### C.1    Gating-only ablation

With the comparator removed,

$$\frac{dw_i}{dt} = y(t)x_i(t).$$

For an isolated pair, the induced window is

$$\Delta w_{\text{gate}}(\Delta t) = \begin{cases} \dfrac{e^{-r_E \Delta t}}{r_A + r_E}, & \Delta t \geq 0, \\[2ex] \dfrac{e^{r_A \Delta t}}{r_A + r_E}, & \Delta t < 0. \end{cases}$$

This expression is always nonnegative. It is therefore a coincidence detector but not an STDP rule with both potentiation and depression. Its total integrated area is

$$A_{\text{gate}} = \frac{1}{r_A r_E}.$$

For the fitted rates, this is approximately 72.435 in the corresponding unscaled area units.

### C.2    Comparator-only ablation

With the overlap gate removed,

$$\frac{dw_i}{dt} = y(t) - x_i(t).$$

For a full-tail isolated pair containing one presynaptic and one postsynaptic event, the integrated contribution is the sum of the two independent trace integrals,

$$\Delta w_{\text{comp}} = \int y(t)\, dt - \int x_i(t)\, dt = \frac{1}{r_A} - \frac{1}{r_E}.$$

With the fitted rates this equals approximately 7.2915. The value depends on the trace time constants, but not on the relative spike timing once both full tails are included. Thus the comparator alone does not produce a timing-localized biphasic STDP window.

### C.3 Summary of ablation outcomes

Table 6: Component ablation summary. Variants are written in the revised trace notation, omitting the common scale $\eta$.

| Variant | Outcome | Interpretation |
|---------|---------|----------------|
| $yx_i$ | Potentiation-only | Captures overlap/coincidence but cannot generate LTD. |
| $y - x_i$ | Not overlap-gated; full-tail pair integral independent of lag | Supplies a sign comparison but not timing-localized STDP. |
| $x_i(1 - x_i)$ | Potentiation-only | Saturated postsynaptic trace removes bidirectionality. |
| $y(y - 1)$ | Depression-only | Saturated presynaptic trace removes bidirectionality. |
| $-x_i^2$ | Depression-only | Removing the postsynaptic factor eliminates coincidence-dependent potentiation. |
| $y^2$ | Potentiation-only | Removing the presynaptic factor eliminates coincidence-dependent depression. |
| $yx_i(y - x_i)$ | Biphasic and timing-sensitive | Combines local overlap gating with temporal comparison. |

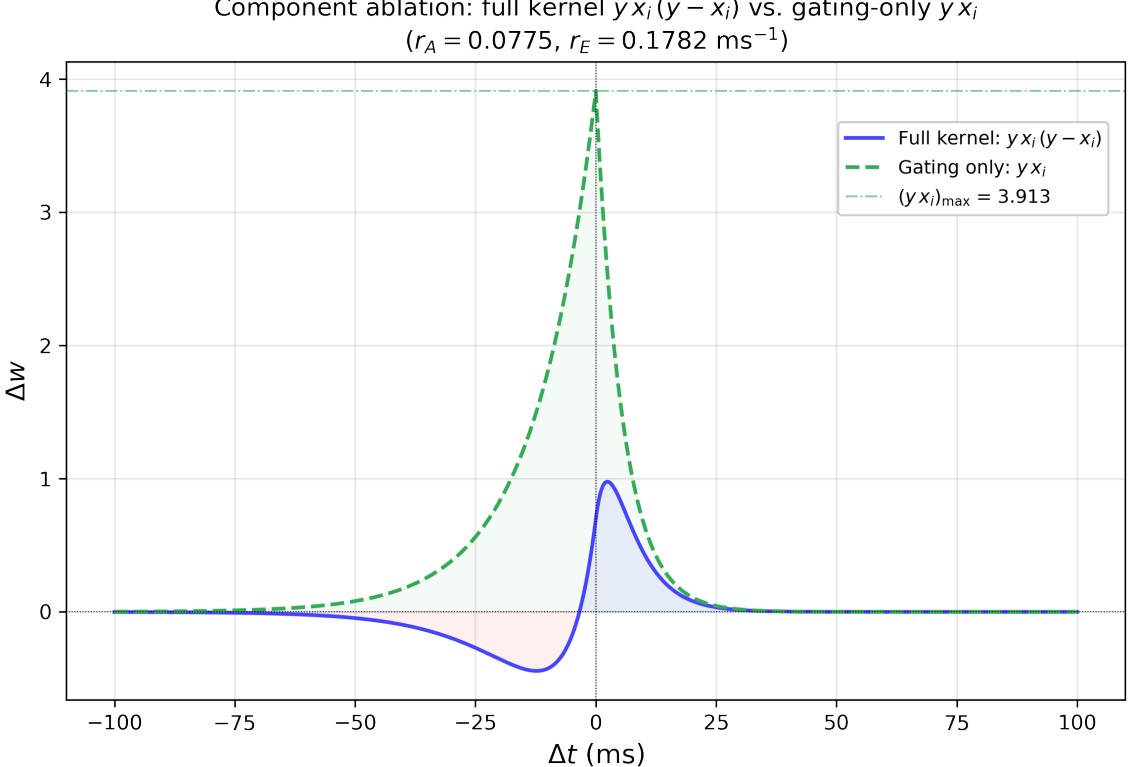

Figure 12: Component ablation comparing the full kernel $y\,x_i\,(y - x_i)$ (blue solid) against the gating-only variant $y\,x_i$ (green dashed), using the fitted decay rates $r_A = 0.0775\,\mathrm{ms}^{-1}$, $r_E = 0.1782\,\mathrm{ms}^{-1}$. The gating-only window is strictly non-negative: it detects spike coincidence but cannot produce depression, and its total integrated area is $A_{\mathrm{gate}} = 1/(r_A r_E) \approx 72.4$. The full kernel restores biphasic timing sensitivity by multiplying the overlap gate with the trace comparator $(y - x_i)$, which flips the sign of the update when the presynaptic trace dominates. Shaded regions highlight the potentiation (blue) and depression (red) lobes of the full kernel. The dash-dotted horizontal line marks the gating-only peak at $\Delta w \approx 3.91$.

The ablations should be read as design-rationale evidence for the isolated-pair kernel. They do not imply that the unnormalized additive rule is stable by itself. As emphasized in the main text, stability and competition arise from the normalization mechanism in Section 4.

## D  Constrained Fit: Zero Crossing at $\Delta t = 0$

To assess whether the near-coincident positive-update regime (Section 3.3) reflects structure in the data or is merely imposed by the continuous kernel, we compared a free fit (where $r_A$ and $r_E$ are independently optimized) against a constrained fit that forces the zero crossing to occur exactly at $\Delta t = 0$ by requiring $r_A = r_E$. Both fits minimize sum-of-squared differences to the Bi & Poo (1998) measurements using the same digitized data and the same closed-form window (Equations 11–13), differing only in the constraint on the decay-rate ratio.

The free fit uses the parameters reported in Section 3.1 ($r_A = 0.0775$ ms$^{-1}$, $r_E = 0.1782$ ms$^{-1}$, $R^2 = 0.63$). The constrained fit achieves $R^2 = 0.38$ with $r = 0.087$ ms$^{-1}$, a decrease of $\Delta R^2 = 0.25$. Model comparison statistics favor the free fit: $\Delta\text{AIC} = -19.0$ and $\Delta\text{BIC} = -17.3$ (negative values favor the model with lower information loss, which here is the free fit despite having one additional parameter).

Figure 13a shows both fitted curves overlaid on the raw data. Figure 13b shows the residuals in the near-zero interval $[-10, 10]$ ms. The constrained fit produces larger residuals precisely in the region where the free fit's shifted zero crossing captures the positive raw observations at small negative lags.

This analysis does not prove a distinct biological mechanism at negative lags. It does show that forcing the classical sign change at $\Delta t = 0$ incurs a measurable and substantial cost to fit quality ($\Delta R^2 = 0.25$, $\Delta\text{AIC} = -19.0$). The free zero crossing is not a coincidence of the optimizer; it reflects structure in the data that the constrained model cannot accommodate.

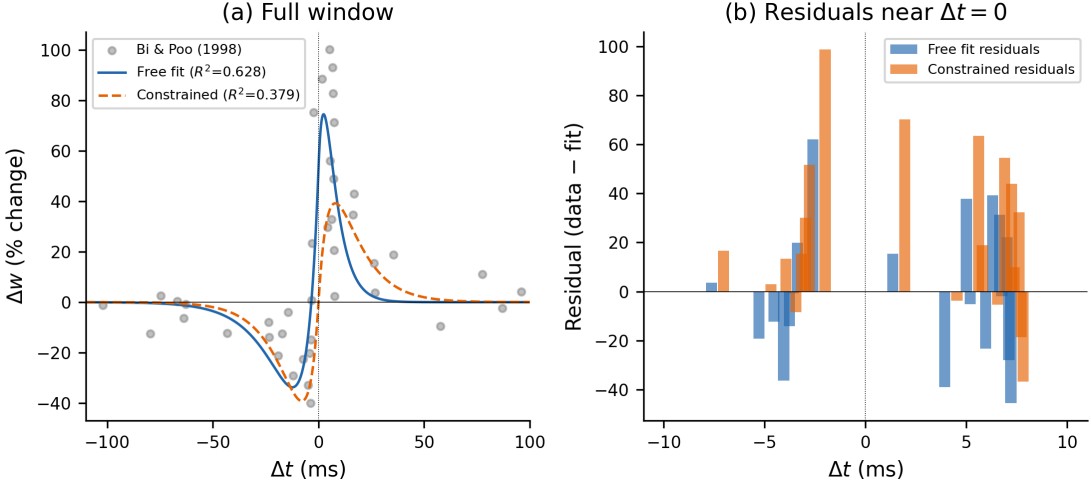

Figure 13: Free vs. constrained fit to the Bi and Poo (1998) data. **(a)** Full STDP window. The free fit (blue, $R^2 = 0.63$) uses the parameters from Section 3.1; the constrained fit (orange dashed, $R^2 = 0.38$) forces $r_A = r_E$, placing the zero crossing at $\Delta t = 0$. **(b)** Residuals in the near-zero interval $[-10, 10]$ ms. The constrained fit produces systematically larger residuals at small negative lags where the raw data are positive.

