# OpenReview forum: "STDP as Probabilistic Attribution: An Exact-Balance Continuous Kernel for Normalized Temporal Credit Assignment"
_TMLR — Under review for TMLR_

### Review · Reviewer_Tj7J · 2026-07-01

**Summary Of Contributions:**

The authors introduce a new mathematical model of spike timing plasticity. In contrast to prior work, they do not specify the causal and acausal branches separately, but describe weight changes as a differential equation from which cumulative weight changes for a spike pairs ("STDP windows") can be derived. This differential equation depends on exponentially decaying traces of the most recent pre- and postsynaptic spikes, respectively. Consequently their model captures "nearest neighbor" interactions, i.e., only contributions from isolated pairs of pre- and postsynaptic spikes. As a main result they show analytically that for uniformly distributed spike intervals, the cumulative weight change is zero. They then fit their model to the original Bi & Poo data and find a positive weight changes for coincident spikes. Finally, they investigate a modified version of their model with added normalization numerically in an "open-loop" setting and demonstrate that it successfully learns the relative event-rate of afferents.

*Key strengths*
- The definition of the kernel via a single function of pre- and postsynaptic traces is quite appealing.
- The simulations use exact integration techniques.
- The authors are precise about what they _don't_ claim.

*Key weaknesses*
- Nearest-neighbor spike pairing, while analytically convenient, is biologically questionable.
- The "main result" could also be interpreted as a critical weakness: the absence of separate LTP/LTD amplitudes reduces flexibility of the model; for example, maybe one would like the expected weight change to be negative, to mimic weight decay?
- The assumptions of main result are questionable: presynaptic activity drives postsynaptic activity, so  uniform interval distributions (uncorrelated pre- and postsynaptic activity) would be only expected for zero weight?
- The experimental fit is based purely on the original Bi & Poo data, yet the authors make significant claims about "near synchronous attributions". A brief search reveals a plethora of more recent experimental results which could help interpret these results and broaden their scope, see for example references in
  - Brzosko, Z., Mierau, S. B., & Paulsen, O. (2019). Neuromodulation of spike-timing-dependent plasticity: past, present, and future. Neuron, 103(4), 563-581
  - Debanne, D., & Inglebert, Y. (2023). Spike timing-dependent plasticity and memory. Current opinion in neurobiology, 80, 102707
- The open-loop setting (presynaptic spikes independently trigger postsynaptic spikes) is quite artificial, and it remains unclear how the obtained insights generalize. Furthermore, these results crucially rely on additional ingredients _not_ part of the original model (normalization). This makes it hard(er) to evaluate the contribution of the original model.
- A problem across Sec. 5 ("verification"): all empirical results are compared to the mean-field result. What are these simulations testing besides which experimental conditions best match the assumptions entering the mean-field derivation?

**Audience:**

Yes

**Audience Explanation:**

I cannot possibly claim that no individuals would be interested in the findings of the paper. However, in the authors' own words, besides the mathematical description of a new STDP kernel, the results seem to mostly constitute "unit tests" and "theory-level numerical checks" (of their specific condition and implementation), while "functional benchmarks [...] are future work".

**Broader Impact Concerns:**

No concerns.

**Claims And Evidence:**

Yes

**Claims Explanation:**

The math is sound, the numerical experiments are appropriate for the claims, and the authors communicate limitations clearly.

**Requested Changes:**

*Critical for recommendation*
- I feel like the manuscript could be shorter, and would encourage the authors to streamline the content. Which theoretical, analytical and numerical results form the core contributions and which are (merely) validation/"verification"?
- Related to the previous point, I think the language and presentation could be improved. Some formulations seem odd, e.g., "temporal attribution scope", "biphasic ODE", but maybe I'm just not familiar with the corresponding literature.
- The authors should weaken claims about their experimental fit or include additional datasets.
- My intuition fails in section 3.5 for the background drift, maybe the authors can clarify. I understand that isolated spike pairs introduce no weight change for uniformly distributed intervals. However, for dense spiking this is apparently not true (if I interpret eq40 correctly?). Why can't one interpret the weight change in the dense setting as a sum of the truncated windows of isolated pairs? If the truncation is random, shouldn't the expectation value of their sum be zero?
- The authors should clarify why weight clipping is necessary in simulations.
- Apply new rule to some closed loop functional task and compare performance to (selected) existing models.

*Strengthen*
- Clarify how this new model would implement anti-stdp. Could this simply be achieved via a negative learning rate?
- Relation/comparison to Oja's rule, which has normalization built in ($\dot w \sim y(x - yw)$).

---

### Review · Reviewer_Vvzs · 2026-07-14

**Summary Of Contributions:**

The submission proposes a continuous, differentiable STDP rule based on cubic trace interactions that can be analyzed in closed form.

Main contributions are that the submission:

1. Showed that the kernel is potentiation-depression balanced for an isolated pair of spikes
2. Evidence that the proposed kernel can fit experimental data under certain parametrization
3. Derived analytical update and fixed point for mean-field weight dynamics under a certain post-synaptic spikes generative model, and, with assumptions, interpreted the fixed point form as probabilistic attribution.
4. Provided simulations examining convergence of the proposed model under various data generating regimes and comparisons with several progressively matched classical STDP baselines.

Key strengths

The STDP kernel proposed by the submission unifies overlap detection and sign selection. It is smooth and easier to analyze as an analytical object. The analytical results are largely relevant and interesting (for example it is nice that the balanced result doesn’t require extra tuning). The submission is also very explicit about its scope and limitations.

Key weaknesses

The proposed probabilistic attribution interpretation is 1) a result of normalization and largely unrelated to the exact form of the kernel 2) requires very strong assumptions on the generative process of the post-synaptic spikes. The claim that the proposed kernel *outperforms* various classical STDP kernels in convergence speed is in a very narrow sense (will elaborate in later sections). The submission appears to be heavily LLM edited (often to limit the scope?) which made certain sections difficult to read. The LLM edits dilute the central points and are counterproductive to keeping the argument well structured.

**Audience:**

Yes

**Audience Explanation:**

The submission would be interesting to researchers studying STDP rules and other synaptic updates.

**Broader Impact Concerns:**

I have no concerns related to broader impact.

**Claims And Evidence:**

No

**Claims Explanation:**

The first two contributions are well supported.

The probabilistic attribution framing on the other hand relied on strong assumptions. For example, as section 4.5 pointed out, it would not hold when there is some level of baseline firing at the post-synaptic neurons. The open-loopness of this weight quantity (since $w$ doesn’t contribute to the data generating process) also makes the consequence of $w$ being interpreted as the posterior of event attribution somewhat unclear. The mean-field fixed point from equation (34) is also independent of K, so the claim in the abstract that the work “introduces a unified kernel… that connects… updates to probabilistic attribution” seems too strong.

Section 5 evaluates the convergence properties of the proposed weight dynamics to the fixed point derived in 4.4. Firstly, I do think that the mean field and the associated equilibrium analysis is valid, and that the simulation results shows convergences under sparse regime which is in accordance with theory. In addition, it is nice that section 5.5 compares the proposed kernel with progressively better matched STDP including ones with K is matched.

However, since the post data in the simulation is generated exactly at 5ms following pre, the convergence speed of each algorithm is roughly governed by $O(1/K)$ where $K$ is dependent on $W(5ms)$, and the limiting MSE is related to the variance of the update for different spike pairs $\int W(\Delta t)^2 d\Delta t$. My understanding is that against the most controlled baseline, the proposed kernel converges better in terms of limiting MSE mostly because of a reduced update variance (which is not matched). And it seems like all the updates without hard-reset might NOT converge to the same fixed point in 4.4. Again, due to the open-loopness, it is unclear why fast convergence to this particular fixed point is superior. Furthermore, under most settings we can not expect the weights to converge to the probabilisitic attribution fixed point. For example under the heterogeneous causal delay setting, even if we assume the exclusive and exhaustive partition, there’s no clear interpretation of the kernel weighted fixed point. So $w^*$ is difficult to interpret even as a tracking metric in open loop setting. Therefore, I am not convinced that “the proposed kernel outperforms classical STDP baselines”.

**Requested Changes:**

Here are the more specific things that I would like to discuss with the authors or that have confused me during reading.

Abstract seems very verbose and contained LLM style sentences like “Our contribution is therefore not a claim of asymptotic speedup…” which distracts the readers.

Generally, it would be very helpful if there are less of the LLM edits (for example ”Not X but Y”, “should be read as X rather than Y”) throughout the paper. I think it is sufficient to justify the decisions made in the paper instead of relying on heavy use of negative scope limiting statements.

Section 3.6

1. At the end the authors mentioned “the main claim that the network-level stability is supplied by normalization”, is this referring to the results in section 4?

Section 4.1

1. My understanding is that $q_i = P(Post | Pre_i)$ here describes the generative process, therefore associated with p^{excess}, it is not the generic $q_i = P(Post | Pre_i)$ of just any post spike. It is unclear from this section that is the case with this notation.
2. The discussion of the K approximation could be moved to 4.3, it was confusing in 4.1

Section 4.2

1. This section generally assumed that structured access events $p_i^{excess}$ only has mass where $W(\Delta t) > 0$ (equation (28)). While it seems reasonable to only model the causal aspects of the excess component, structured negative-lag excess could also exist especially in recurrent networks (see Ostojic, Brunel, & Hakim 2009). The $q_i$-based attribution derivation won’t model the negative excess since they can not be normalized into a posterior. Modeling structured negative excess might be useful in explaining the near coincidence events.
2. Consider adding the point process definition of the pair-rate function G.

Section 4.4

1. In Eqn (37), multiple input could have spiked before a particular post event even under mutually exclusive and exhaustive partition (it just didn’t generate the conditioned post event). $w_i^*$ here seems to track the probability that an attributed postsynaptic event was generated by a presynaptic event from source $i$. Perhaps the notation $P(Pre_i|Post)$ could be replaced by P(C=i | an attrubuted post occurred) where C denote the $i$th source neuron.

Section 5

1. Figure 2c, 4b, 5b, 6b, 8b looks like the log-uniform sampling scheme has produced a lot of samples in the small weight regime that makes the evaluation of correlation coefficient in the whole data range difficult. I was wondering whether a more uniform sampling scheme will make difference between different methods more pronounced or narrower.
2. In figure 7b where the target weight is more uniform, the converged final weights seems to be linearly related to the target weight but it doesn’t seem to have converged to the target weights (systematically away from the diagonal). I was wondering why this might be.
3. Generally I was wondering why the data generating process is set up to be log uniform instead of something that allows entries of $w^*$ to be uniform.

---

### Review · Reviewer_fr1z · 2026-07-21

**Summary Of Contributions:**

The paper studies the cubic spike timing-dependent plasticity rule

$$
\dot w_i=\eta y x_i(y-x_i),
$$

where $x_i$ and $y$ are exponentially decaying pre and postsynaptic traces
with nearest-spike hard resets. The product $yx_i$ gates plasticity, while
$y-x_i$ determines its sign. The paper makes three main contributions:

1. **Isolated-pair window.** The rule produces a continuous biphasic STDP
   window without separate LTP and LTD branches.
2. **Exact balance.** The signed area of this window is exactly zero for all
   decay rates $r_A,r_E>0$.
3. **Normalized fixed point.** Under a sparse mean-field approximation,
   positive attribution increments followed by multiplicative afferent
   normalization give
   $w_i^*=\nu_iq_i/\sum_j\nu_jq_j$, a normalized kernel-weighted attribution
   target.

**Strengths.** The single cubic rule and its exact-balance property are interesting. The paper is well organized, the main derivations are
correct, and the limits of the results are stated clearly.

**Weaknesses.** The title gives more prominence to the posterior interpretation
than the result supports. The strongest result is
normalized kernel-weighted attribution; a Bayesian posterior interpretation requires additional
event-partition assumptions. The best classical baseline also matches only one
symmetric timescale, not the two lobe shapes separately. A few definitions and
statistical comparisons could be made more precise. These issues do not affect
the main mathematical results.

**Audience:**

Yes

**Audience Explanation:**

A single cubic interaction that produces a biphasic, exactly balanced STDP
window without separately tuned LTP/LTD amplitudes is an interesting result.
It should interest the computational-neuroscience part of TMLR's audience,
especially readers working on plasticity kernels.

**Broader Impact Concerns:**

None. This is a theoretical and computational study of a plasticity rule.

**Claims And Evidence:**

Yes

**Claims Explanation:**

I checked the main closed-form expressions independently; they are mathematically consistent under the stated
approximations.

The paper is also explicit about its scope: the exact balance applies to isolated
pairs, the dense firing can create residual drift $B_i$, the posterior interpretation needs
extra assumptions, and the experiments provide targeted checks of their theory.

Two claims should be framed more carefully:

- The title emphasizes probabilistic attribution. The abstract already explains that the posterior-probability interpretation is valid only under additional assumptions, but the result is normalized
  kernel-weighted attribution. The identity with
  $P(\mathrm{Pre}_i\mid\mathrm{Post})$ requires a mutually exclusive and
  exhaustive partition of postsynaptic events, which independent afferents do
  not generally provide.
- The best classical baseline uses one symmetric timescale,
  $\tau=1/\sqrt{r_Ar_E}$. It therefore does not separately match the widths or
  moments of the asymmetric positive and negative lobes. The remaining
  performance gap cannot yet be attributed cleanly to the cubic form alone.

**Requested Changes:**

The first two changes would most improve the paper's accuracy and clarity. The
others would strengthen the evidence. None of them is critical for acceptance.


1. **Make the title match the result.** The abstract
   already explains the required assumptions, but the title can still suggest
   a general posterior interpretation. Consider presenting
   $P(\mathrm{Pre}_i\mid\mathrm{Post})$ as a special case.
2. **Clarify the remaining definitions.** Section 2 already provides the trace
   ODEs and hard-reset rule. It would be useful to state the units of $w_i$ and $\eta$,
   specify how coincident pre/post events are processed, distinguish spikes from
   attribution events, and define the event variable for
   $P(\mathrm{Post}\mid\mathrm{Pre}_i)$.

**Would strengthen the paper:**

3. **Use a more closely matched baseline.** Compare against a two-timescale
   classical STDP window that separately matches the positive and negative
   lobes while preserving area balance and the update at 5 ms. Since all
   methods use the same event streams, report paired-seed differences with
   confidence intervals.
4. **Separate zero-crossing location from lobe asymmetry.** The constraint
   $r_A=r_E$ changes both properties, so the worse constrained fit cannot be
   attributed only to moving the crossing to zero. Either compare with a model
   that retains asymmetric lobes while crossing at zero or narrow this claim.
5. **Test the explanation for residual error.** Compare the simulations with
   $$
   w_i^{*,B}=\frac{KR_i+B_i}{KR_{\mathrm{tot}}+\sum_k B_k},
   $$
   and report results with and without clipping. This would show whether the
   remaining error comes from background drift or the nonnegativity boundary.